# $D^2$MATCH: LEVERAGING DEEP LEARNING AND DEGENERACY FOR SUBGRAPH MATCHING

## ABSTRACT

Subgraph matching is a fundamental building block for many graph-based applications and is challenging due to its high-order combinatorial nature. However, previous methods usually tackle it by combinatorial optimization or representation learning and suffer from exponential computational cost or matching without theoretical guarantees. In this paper, we develop $D^2$Match by leveraging the efficiency of Deep learning and Degeneracy for subgraph matching. More specifically, we prove that subgraph matching can degenerate to subtree matching, and subsequently is equivalent to finding a perfect matching on a bipartite graph. This matching procedure can be implemented by the built-in tree-structured aggregation mechanism on graph neural networks, which yields linear time complexity. Moreover, circle structures, abstracted as *supernodes*, and node attributes can be easily incorporated in $D^2$Match to boost the matching. Finally, we conduct extensive experiments to show the superior performance of our $D^2$Match and confirm that our $D^2$Match indeed tries to exploit the subtrees and differs from existing learning-based subgraph matching methods that depend on memorizing the data distribution divergence.

## 1 INTRODUCTION

Graphs serve as a common language for modeling a wide range of applications (Georgousis et al., 2021) because of their superior performance in abstracting representations for complex structures. Notably, subgraph isomorphism is a critical yet particularly challenging graph-related task, a.k.a., subgraph matching at the node level (McCreesh et al., 2018). Subgraph matching aims to determine whether a query graph is isomorphic to a subgraph of a large target graph. It is an essential building block for many applications, as it can be used for alignment (Chen et al., 2020), canonicalization (Zhou & Torre, 2009), motif matching (Milo et al., 2002; Peng et al., 2020), etc.

Previous work tries to resolve subgraph matching in two main streams, i.e., combinatorial optimization (CO)-based and learning-based methods (Vesselinova et al., 2020). Early algorithms often formulate subgraph matching as a CO problem that aims to find all exact matches in a target graph. Unfortunately, this yields an NP-complete issue (Ullmann, 1976; Cordella et al., 2004) and suffers from exponential time cost. To alleviate the computational cost, researchers have employed approximate techniques to seek inexact solutions (Mongiovì et al., 2010; Yan et al., 2005; Shang et al., 2008). An alternative solution is to frame subgraph matching as a machine learning problem (Bai et al., 2019; Rex et al., 2020; Bai et al., 2020) by computing the similarity of the learned representations at the node or graph levels from two graphs. Though learning-based models can attain a solution in polynomial time, they provide little theoretical guarantee, making the results suboptimal and lacking interpretability. If not worse, the learning-based methods often cannot obtain the exact match subgraphs.

Ideally, we hope to develop a subgraph matching algorithm that can leverage the efficiency of learning methods while still maintaining theoretical guarantees. We approach this by building the connection between subgraph matching and perfect matching on a bipartite graph. We prove that finding the corresponding nodes between the query graph and the target one is equivalent to checking whether there is a perfect matching on the bipartite graphs generated by the nodes from the query graph and the target one recursively, yielding a much more efficient subgraph matching algorithm solved in polynomial time.

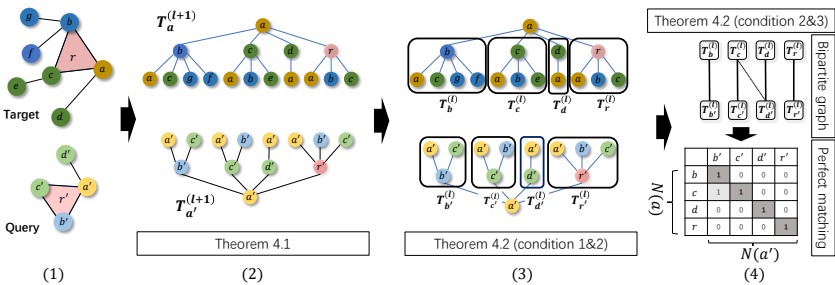

Figure 1: An illustration of the proposed degeneracy procedure for subgraph matching. Step (1) & (2) are to determine the isomorphism of a pair $(a, a')$ by examining whether their corresponding subtrees, i.e., $T_a^{(l+1)}$ and $T_{a'}^{(l+1)}$, are subtree isomorphic based on Theorem 1. Step (2) & (3) are to simplify the determination of $T_{a'}^{(l+1)} \subset T_a^{(l+1)}$ by checking whether a subtree-isomorphism holds for every $l$-depth rooted subtree in $N(a')$ to a unique $l$-depth subtree in $N(a)$ based on Theorem 2. Step (3) & (4) are to transform the subtree-isomorphism relation between $N(a')$ and $N(a)$ to edges in a bipartite graph and interpret the problem as a perfect matching on a bipartite graph.

This degeneracy allows us to harness the power of Graph Neural Networks (GNNs) to fulfill the matching by deploying a built-in tree-structured aggregation mechanism in GNNs. Operating on node-level correspondences offers a node matching matrix, which allows us to locate the matched subgraph directly. To incorporate more information, we augment the bipartite graph with *supernodes*, which wraps the circles, into the perfect matching procedure; see Fig. 1 for an illustration of the basic idea. Moreover, node attributes can be easily included accordingly.

Our primary contribution is three-fold: (1) $D^2$Match proposes a novel learning-based subgraph matching method, which frames the subgraph matching problem as perfect matching on a bipartite graph. (2) We theoretically prove that this matching procedure can be implemented by the built-in tree structured aggregation mechanism on GNNs and yields linear time complexity. Moreover, we can easily incorporate circle structures, abstracted as *supernodes*, and node attributes into our $D^2$Match to boost the performance. (3) Extensive empirical evaluations show that $D^2$Match outperforms state-of-the-art subgraph matching methods by a substantial margin and uncover that learning-based methods tend to capture the data distribution divergence rather than performing matching.

## 2 RELATED WORK

Subgraph matching is to check whether a query graph is subgraph isomorphic to the target one (McCreesh et al., 2018). Here, we highlight three main lines of related work:

**Combinatorial optimization (CO)-based methods** first tackle subgraph matching by only modeling graph structure (Ullmann, 1976). Recent work starts to facilitate both graph structure and node attributes (Han et al., 2013; Shang et al., 2008). These combinatorial optimization methods often rely on backtracking (Priestley & Ward, 1994), i.e., heuristically performing matching on each pair of nodes from the query and the target graphs. Such methods suffer from exponential computing costs. A mitigated solution is to employ an inexact matching strategy. Early methods first define metrics to measure the similarity between the query graph and the target graph. Successive algorithms follow this strategy and propose more complex metrics. For example, Mongiovì et al. (2010) convert the graph matching problem into a set-cover problem to attain a polynomial complexity solution. Yan et al. (2005) introduce a thresholding method to filter out mismatched graphs. Khan et al. (2011) define a metric based on neighborhood similarity and employ an information propagation model to find similar graphs. Kosinov & Caelli (2002) and Caelli & Kosinov (2004) align the nodes' eigenspace and project them to the eigenspace via clustering for matching. However, most of these algorithms cannot scale to large graphs due to the high computational cost, and their hand-crafted features make them hard to generalize to complex tasks.

**Learning-based methods** typically compute the similarity between the query and target graphs, e.g., comparing their embedding vectors. Bai et al. (2019) adopt GNNs to learn node representations of the query and target graphs, which employs a neural tensor network to match the graph pairs. One immediate challenge is that a single graph embedding vector cannot capture the partial order information for subgraph matching. Thus, Rex et al. (2020) train a GNN model to represent

graphs while incorporating order embeddings to learn the partial order. These methods can compute graph-level representations, achieving high computational efficiency. However, they miss the node-level information, which may lose critical details in subgraph matching. To perform node-level matching, several methods (Bai et al., 2020; Li et al., 2019) introduce the graph-level representation into the node-level matching problem. These methods often adopt different attention mechanisms to generate pairwise relations. However, abusing the attention mechanism makes the model lack interpretability and theoretical guarantee. Others transform the subgraph matching problem into an edge matching problem and generate prediction results through the matching matrix obtained by Sinkhorn's algorithm (Roy et al., 2022), thereby providing interpretability for the model. The process of turning node matching into edge matching, however, loses necessary information about edges' relation, such as edges' common nodes, which hurts the expressibility of the model.

**Graph Neural Networks (GNNs)** are powerful techniques (Xu et al., 2019; Kipf & Welling, 2017) yielding breakthroughs in many key applications (Hamilton et al., 2017b). Over the last years, there has been considerable progress in proposing different ways of aggregating. For example, Graph-SAGE (Hamilton et al., 2017b) aggregates nodes features with mean/max/LSTM pooled neighboring information. Graph Attention Network (GAT) (Velickovic et al., 2018) aggregates neighbor information using learnable attention weights. Graph Isomorphism Network (GIN) (Xu et al., 2019) converts aggregation as a learnable function based on the Weisfeiler-Lehman (WL) test instead of prefixed ones as other GNNs, aiming to maximize the performance of GNNs. However, the WL test (Xu et al., 2019) cannot address the subgraph matching problem because it hashes the tree structure and loses the partial order information of subgraph matching.

## 3    PRELIMINARY

To make the notation consistent, we define them as follows: Let $A_\mathcal{Q}$ and $A_\mathcal{T}$ be the adjacency matrix of the query graph $G_\mathcal{Q}$ and the target graph $G_\mathcal{T}$, respectively. $N(\cdot)$ denotes the neighbor set of a given node. $|\cdot|$ denotes the size of a set. $T_v^{(l)}$ defines the subtree whose root is $v$ and expands up to $l$-hop neighbors of $v$ or $l$-layer of the subtree. In the paper, the concepts of the $l$-hop neighbors and the $l$-layer subtrees are interchangeable.

**Problem Definition:** Suppose we are given a query graph, $G_\mathcal{Q}(V_\mathcal{Q}, E_\mathcal{Q})$, and a target graph, $G_\mathcal{T}(V_\mathcal{T}, E_\mathcal{T})$. Here, $(V_\mathcal{Q}, E_\mathcal{Q})$ and $(V_\mathcal{T}, E_\mathcal{T})$ are the pairs of vertices and edges related to the query graph and the target graph, respectively. Besides, the node attributes of the query graph and the target graph are denoted as $X_\mathcal{Q} \in \mathbb{R}^{|V_\mathcal{Q}| \times D}$ and $X_\mathcal{T} \in \mathbb{R}^{|V_\mathcal{T}| \times D}$, respectively, where $D$ is the size of the node attributes.

The problem of subgraph matching is to identify whether the query graph is subgraph isomorphic to the target graph, i.e. if there exists an injective $f : V_\mathcal{Q} \to V_\mathcal{T}$ such that $\forall u, v \in V_\mathcal{Q}, (u, v) \in E_\mathcal{Q} \Leftrightarrow (f(u), f(v)) \in E_\mathcal{T}$. Without loss of generality, we hypothesize that $G_\mathcal{Q}$ is the subgraph of $G_\mathcal{T}$, i.e., $G_\mathcal{Q} \subset G_\mathcal{T}$ with $|V_\mathcal{Q}| < |V_\mathcal{T}|$. In essence, the subgraph isomorphism test is to check a matching matrix $S \in \{0, 1\}^{|V_\mathcal{T}| \times |V_\mathcal{Q}|}$, where $S_{ij} = 1$ if and only if node pair$(i, j)$ is matched. $G_\mathcal{Q}$ is isomorphic to $G_\mathcal{T}$ is equivalent to checking whether the following conditions hold:

$$\sum_{i=1}^{|V_\mathcal{T}|} S_{ij} = 1, \quad \sum_{j=1}^{|V_\mathcal{Q}|} S_{ij} \le 1 \tag{1}$$

In the following, we first define several key concepts in our work.

**WL Subtree:** The Weisfeiler-Lehman (WL) test is an approximate solution to the graph isomorphism problem with linear computational complexity (Shervashidze et al., 2011). The WL test performs the aggregation on nodes' labels and their neighborhoods recursively, followed by hashing the aggregated results into unique new labels. As a result, this test produces an unordered tree for each node, called the WL subtree, which is a balanced tree with the height of the number of iterations. After repeating the algorithm $k$ times, the obtained WL subtree for a node includes the structural information of the $k$-hop subgraph from that node. Research shows that the expressiveness of the WL subtree is the upper limit of message passing GNNs (Xu et al., 2019).

**Subtree Generation:** Considering a node $v$, we can obtain a subgraph $Sub_v^{(l)}$ by taking the $l$-hop neighbor of $v$. Given any tree generation method, e.g., the WL subtree, we always obtain a corresponding subtree whose root is $v$:

$$T_v^{(l)} = \Psi(Sub_v^{(l)}), \tag{2}$$

where $\Psi$ is a subtree generation function. Unless stated otherwise, we employ the WL subtree to generate subtrees for a given node due to its uniqueness (Xu et al., 2018). Instead of explicitly constructing such trees, we can run GNNs in a graph, since building a $k$-order WL subtree is equivalent to aggregating $k$ times in GNNs (Xu et al., 2018). Notice that traditional methods such as Breadth-First-Search (BFS) and Depth-First-Search (DFS) are not applicable at this work because they do not satisfy the uniqueness property. In particular, the tree generated for the same node by BFS or DFS will be different due to different search order.

**Perfect Matching in Bipartite Graphs:** A perfect matching (Gibbons, 1985) is a matching of a graph in which every node of the graph is incident to exactly one edge. Performing perfect matching on a bipartite graph can be solved according to Hall's marriage theorem (Hall, 1935).

**Theorem 3.1.** (Hall's marriage theorem) Given a bipartite graph, $B(X, Y, E)$ that has two partitions: $X$ and $Y$ and $|Y| \leq |X|$, where $E$ denotes the edges. The necessary and sufficient condition of the existence of the perfect matching in $B(X, Y, E)$ is : $\forall W \subseteq Y, |W| \leq |N(W)|$, where $N(W)$ is the neighborhood of $W$ defined by $N(W) = \{b_j \in X : \exists a_i \in W, (a_i, b_j) \in E\}$.

## 4 THE PROPOSED METHOD

This section describes the proposed $D^2$Match. We first introduce the degeneracy of subgraph matching and propose an aggregation-based operation to address the degenerated problem, an efficient solution with linear time complexity. Following, we introduce two components designed to strengthen the matching ability by incorporating the circle structure and node attributes.

### 4.1 ON THE DEGENERACY OF THE SUBGRAPH MATCHING PROBLEM

We approach the subgraph matching problem from a degeneracy perspective, framing this problem as a subtree matching problem with linear complexity. A fundamental question to the subgraph matching problem is on what conditions one subgraph is isomorphic to the other. Since the subgraph matching problem is NP-complete, the exact answer to this question becomes impractical. Instead, we can reduce the answer of finding both sufficient and necessary conditions to that of necessary only. What follows is to construct a criterion that any isomorphic pairs can meet. We know that the subtree matching problem yields polynomial time cost. Inspired by this, we attempt to construct the criterion by taking advantage of the subtrees rooted at these nodes, which degenerate subgraph matching to subtree matching and are guaranteed by the following theorem.

**Theorem 4.1.** Given a target graph $G_{\mathcal{T}}(V_{\mathcal{T}}, E_{\mathcal{T}})$ and a query graph $G_{\mathcal{Q}}(V_{\mathcal{Q}}, E_{\mathcal{Q}})$, if $G_{\mathcal{Q}} \subset G_{\mathcal{T}}$, and the subtree generation function $\Psi$ as defined in Eq. (2) meets the following condition:

$$\forall \text{ graph pair } (G_{\mathcal{S}}, G), \text{if } G_{\mathcal{S}} \subset G, \text{then } \Psi(G_{\mathcal{S}}) \subset \Psi(G), \tag{3}$$

then there exists an injective function $f : V_{\mathcal{Q}} \to V_{\mathcal{T}}$, ensuring the $l$-hop subtrees of the subgraph is isomorphic to the subtrees of the corresponding subgraph:

$$\forall l \geq 1, \mathsf{q} \in V_{\mathcal{Q}}, \mathsf{t} = f(\mathsf{q}) \in V_{\mathcal{T}} \Rightarrow T_{\mathsf{q}}^{(l)} \subset T_{\mathsf{t}}^{(l)}, \tag{4}$$

Please find the proof in Appendix A.1. This theorem provides a necessary condition for the potential isomorphic pairs, i.e., those who pass the test. Given a query graph and a target graph, we can construct an indicator matrix $S \in R^{|V_{\mathcal{T}}| \times |V_{\mathcal{Q}}|}$ by setting $S_{\mathsf{tq}}$ to 1 when $T_q \subset T_t$ and 0 otherwise. The isomorphic test becomes to check the validity of Eq. (1). Due to the favorite property of uniqueness, we employ the WL subtree as the generation function. Thanks to the built-in connection between the WL subtree and GNNs, we can convert the subtree matching problem to the problem of perfect matching on a bipartite graph and derive a GNN-based solution. The following theorem guarantees the conversion:

**Theorem 4.2.** Given a node q in the query graph and a node t in the target graph, the following three conditions are equivalent:

1) $T_{\mathsf{q}}^{(l+1)} \subset T_{\mathsf{t}}^{(l+1)}$.
2) There exists an injective function on the neighborhood of these nodes as $f : N(\mathsf{q}) \to N(\mathsf{t})$, s.t.
   $\forall \mathsf{q}_i \in N(\mathsf{q}), \mathsf{t}_i = f(\mathsf{q}_i), T_{\mathsf{q}_i}^{(l)} \subset T_{\mathsf{t}_i}^{(l)}$.

3) There exists a perfect matching on the bipartite graph $B^{(l)}(N(\text{t}), N(\text{q}), E)$, where $\forall \text{t}_j \in N(\text{t}), \text{q}_i \in N(\text{q}), (\text{t}_j, \text{q}_i) \in E$ if and only if $T_{\text{q}_i}^{(l)} \subset T_{\text{t}_j}^{(l)}$.

The proof is provided in Appendix A.2. The equivalence of the first two conditions implies that matching subtrees of a pair of nodes is equivalent to matching all subtrees from their child nodes. As a result, the indicator matrix needs to be updated recursively. That is, the indicator matrix at the $(l + 1)$-th layer, i.e., $S^{(l)}$, should rely on $S^{(l)}$. Meanwhile, the equivalence of the last two conditions means that matching the subtrees from these child nodes is equivalent to solving the perfect matching on the corresponding bipartite graph whose nodes represent the subtrees of the child nodes. In summary, Theorem 4.2 tells us that subgraph matching is equivalent to delivering perfect matching on a bipartite graph. A visualization of this procedure is shown in Fig. 1.

Motivated by Hall's marriage Theorem 3.1, we develop an efficient algorithm to address the perfect matching procedure. A straightforward solution is to randomly select a subset $W$ from the given set of neighbors, $N(\text{q})$ in $G_{\mathcal{Q}}$, and count whether the corresponding neighbors of $W$ in $B^{(l)}(N(\text{t}), N(\text{q}), E)$, i.e. $N(W)$, have more elements than this subset. After repeating this process multiple times for all node pairs, we obtain a perfect matching when no instance violates the criterion.

*Is it possible to execute all pairs in parallel?* Luckily, we can borrow GNNs to accomplish the perfect matching. Specifically, when computing a perfect matching between node $\text{q} \in G_{\mathcal{Q}}$ and node $\text{t} \in G_{\mathcal{T}}$, one needs to find $W$ such that it satisfies $W \subseteq Y = N(\text{q})$ according to Theorem 3.1. In practice, we can obtain this by sampling the neighbors of node q, equating to sampling the edges, or the Drop Edge operation (Hamilton et al., 2017a). In this way, we obtain a sampled graph $G_{\mathcal{Q}}'$ from the query graph $G_{\mathcal{Q}}$, along with its adjacency matrix $\tilde{A}_{\mathcal{Q}}$. Following Theorem 3.1, we conclude that $W = N'(\text{q})$ with $N'(\text{q}) \subset N(\text{q})$ since node q's neighbors in $G_{\mathcal{Q}}'$ are a subset of the original graph. At each iteration, we will perform the counting w.r.t. $W$ and its neighbor set $N(W)$ for each node pair $(\text{t}, \text{q})$, and check whether $|N(W)| \geq |W|$ holds. To be efficient, we define a binary matrix, $\Phi \in R^{|V_{\mathcal{T}}| \times |V_{\mathcal{Q}}|}$, where its element at $(\text{t}, \text{q})$ corresponds to the result of the node pair $(\text{t}, \text{q})$.

Based on Theorem 4.1, we need to update the indicator matrix $S$ recursively, making the update of $\Phi$ executed in recursion accordingly. We next show that computing $\Phi$ is equivalent to performing the GNN-based aggregation on the related graphs for any given $S^{(l)}$.

**Theorem 4.3.** Given the sampled query graph and the target graph, we can construct their adjacency matrices , $\tilde{A}_{\mathcal{Q}}$ and $A_{\mathcal{T}}$, and the degree matrix of the sampled query graph $\tilde{D}_{\mathcal{Q}} = \text{diag}(\sum_s ((\tilde{A}_{\mathcal{Q}})_{:s}))$. Here, we denote the indicator matrix at the $l$-th hop as $S^{(l)}$. To check the validity of $|N(W)| \geq |W|$ for each node pair, we can check whether each element of $\Phi$ is true or not, where $\Phi := Z_{N(W)} \geq 1$, $Z_{N(W)} = \text{aggregate}_{\text{sum}}(A_{\mathcal{T}}, Z_W^T)$ and $Z_W = \text{aggregate}_{\max}(\tilde{D}_{\mathcal{Q}}^{-1} \cdot \tilde{A}_{\mathcal{Q}}, (S^{(l)})^T)$.

The proof is provided in Appendix A.5. Recalling Theorem 3.1, we need to check $|N(W)| \geq |W|$ for each node pair $(\text{t}, \text{q})$, i.e., to check whether each element of $\Phi$ is true for each sampled $\tilde{A}_{\mathcal{Q}}^{(k)}$. The condition is valid only when $\Phi$ is true for all iterations. Hence we can check the criterion by the following element-wise product:

$$S_{subtree}^{(l+1)} = \bigodot_{k=0}^{K} \Phi^{(l+1)}(\tilde{A}_{\mathcal{Q}}^{(k)}, A_{\mathcal{T}}), \tag{5}$$

where $\bigodot$ denotes the element-wise multiplication between matrices. In practice, $\Phi^{(l+1)}(\tilde{A}_{\mathcal{Q}}^{(k)}, A_{\mathcal{T}})$ considers three cases:

$$\begin{cases} \text{aggregate}_{\text{sum}}(A_{\mathcal{T}}, \text{aggregate}_{\max}(D_{\mathcal{Q}}^{-1} \cdot A_{\mathcal{Q}}, (S^{(l)}))^T) \geq 1 & \text{if} \quad k = 0 \\ \text{aggregate}_{\text{sum}}(A_{\mathcal{T}}, \text{aggregate}_{\max}(\tilde{D}_{\mathcal{Q}}^{-1} \cdot \tilde{A}_{\mathcal{Q}}^{(k)}, (S^{(l)}))^T) \geq 1 & \text{if} \quad k \in [1, K-1] \ , \\ \text{aggregate}_{\min}(A_{\mathcal{Q}}, \text{aggregate}_{\max}(A_{\mathcal{T}}, (S^{(l)})^T)) \geq 1 & \text{if} \quad k = K \end{cases} \tag{6}$$

The above three cases allow us to balance the computation cost and accuracy. Initially, when $k = 0$, we deliver a full-size sampling for all nodes to avoid induction bias. When $k = K$, we perform the single-node sampling such that no node is omitted. The cases of $k \in [1, K-1]$ are computed via downsampling.

We want to highlight the difference between ours and other learning-based methods regarding GNNs. Here we employ a GNN model to accomplish the procedure of subtree matching, along with theo-

retical equivalence. With the subtree representation learning, GNNs let other learning-based models capture the variance of data distribution for similarity inference since deep learning models learn distributional information to distinguish samples from different classes.

## 4.2 BOOSTING THE MATCHING

The last section introduced a new method to address the subgraph matching problem based on the proposed necessary condition, which we call the base model. Without sufficient conditions, it cannot guarantee that all positive isomorphism pairs to be selected precisely. To mitigate this issue, we further incorporate information such as circle structure and node attributes to filter out more non-isomorphism pairs.

### 4.2.1 DEALING WITH CIRCLES

Prior methods often leave the circle structure aside, however, such a structure is often unavoidable and critical in graphs. In particular, learning-based methods rely on the expressibility of GNNs, which cannot model circles due to their subtree-structured aggregation. The underlying idea of our $D^2$Match is to construct the circle structures as *supernodes*, which allow us to formulate the circle matching as a standard subtree matching problem. Before detailing our strategy, we first present two desired properties of the set of circles in a graph.

**Atomic:** Let $r = (v_1, ..., v_{l(r)}, v_1) \in \mathcal{C}$ define a circle and $v(r)$ be the set of nodes of circle $r$, a circle is an *atomic* circle if it does not contain a smaller circle. That is, there is no circle $r'$ such that $v(r') \subset v(r)$. Here, $\mathcal{C}$ is the circle set.

**Consistency:** Each query circle must correspond to one circle in the target graph, i.e., $\exists\ f, \forall r \in \mathcal{C}_\mathcal{Q}, f(r) \in \mathcal{C}_\mathcal{T}$. $\mathcal{C}_\mathcal{Q}$ and $\mathcal{C}_\mathcal{T}$ are the circle sets of the query graph and the target graph, respectively.

The atomic property aims to ensure the compactness of circles, and the consistency attempts to ensure that the relation between a query and a target set of circles is injective. These two properties ensure a well-qualified set for matching. In practice, we can take advantage of *chordless cycles* (West, 2000), to serve our goal of matching circles. We now state our theorem below to show that these cycles satisfy the above consistency and atomic property.

**Theorem 4.4.** Every chordless cycle is atomic. Every chordless cycle $\mathcal{C}_\mathcal{Q}$ in an induced subgraph of the original query graph $G_\mathcal{Q}$ must correspond to a chordless cycle $\mathcal{C}_\mathcal{T}$ in the origin graph $G_\mathcal{T}$.

Please find the proof in Appendix, A.6. This theorem suggests that chordless cycles satisfy the above two properties, making them suitable for representing circles in a graph. To match circles, we introduce an augmented graph by inserting supernodes that embody these circles. Given a length $L$, we can acquire corresponding chordless cycles for the query and target graphs as : $\mathcal{C}_\mathcal{T} = \{l(r) \leq L, r \in CC(G_\mathcal{T})\}, \mathcal{C}_\mathcal{Q} = \{l(r) \leq L, r \in CC(G_\mathcal{Q})\}$. By setting $v_r$ as the supernode of any chordless circle $r$, we connect nodes from the circle $r$ to this supernode, resulting in an augmented graph. Note that supernodes can only match other supernodes to keep the matching of non-circles untainted. To this end, we transform the circle matching as the subtree matching problem such that we can employ the proposed method on the augmented graph directly. Unless otherwise stated, we keep all notations the same in the augmented graph to avoid abusing the notations.

### 4.2.2 DEALING WITH NODES' ATTRIBUTES

Apart from the above structure information, subgraph matching also involves node attributes. Within the context of subgraph matching, learning with node attributes alone may be misled because these cannot catch structural isomorphism. As a result, we employ the obtained subtree indicator matrix to supervise the learning process, aiming to filter out pairs that do not pass the test in the subtree matching. We are thus motivated to enhance the node attributes by concatenating the subtree matching indicator, resulting in the node representation for the query and target graphs as follows:

$$\begin{cases} H_\mathcal{T}^{(l+1)} = GNN_\mathcal{T}^{(l)}(A_\mathcal{T}, \text{concat}(H_\mathcal{T}^{(l)}, MLP(S^{(l)}))) \\ H_\mathcal{Q}^{(l+1)} = GNN_\mathcal{Q}^{(l)}(A_\mathcal{Q}, \text{concat}(H_\mathcal{Q}^{(l)}, MLP(S^{(l)})^T))) \end{cases} \tag{7}$$

Here, we employ an MLP model to reduce the effect of the difference between the node attributes and the indicator matrix, where the latter behaves like a one-hot feature. We concatenate each pair

of representations and then pass it to the MLP to obtain their similarity. For the node pair $(i, j)$, we have the similarity computed as

$$[S_{gnn}^{(l+1)}]_{ij} = MLP(\text{concat}([H_{\mathcal{T}}^{(l)}]_i, [H_{\mathcal{Q}}^{(l)}]_j)). \tag{8}$$

Now we arrive at a generalized indicator matrix that considers both the structure and node attribute information $S^{(l+1)} = S_{gnn}^{(l+1)} \odot S_{subtree}^{(l+1)}$.

---

**Algorithm 1** The $D^2$Match algorithm

---

**Require:** A query graph $G_{\mathcal{Q}}(V_{\mathcal{Q}}, E_{\mathcal{Q}})$ with node attributes $X_{\mathcal{Q}}$, a target graph $G_{\mathcal{T}}(V_{\mathcal{T}}, E_{\mathcal{T}})$ with node attributes $X_{\mathcal{T}}$, iteration number: $L$, sample number: $K$.
**Ensure:** Is $G_{\mathcal{Q}}$ isomorphic to $G_{\mathcal{T}}$
 1: $G_{\mathcal{Q}}(V_{\mathcal{Q}}, E_{\mathcal{Q}}) \leftarrow ChordlessCycleAugment(G_{\mathcal{Q}}); G_{\mathcal{T}}(V_{\mathcal{T}}, E_{\mathcal{T}}) \leftarrow ChordlessCycleAugment(G_{\mathcal{T}});$
 2: $H_{\mathcal{Q}}^{(0)} = X_{\mathcal{Q}}; H_{\mathcal{T}}^{(0)} = X_{\mathcal{T}};$
 3: $S_{subtree}^{(0)} = InitialAssignMatrix(X_{\mathcal{T}}, X_{\mathcal{Q}})$ ▷ Initialize assignment matrix;
 4: **for** $l = 0, 1..., L - 1$ **do**
 5:    **for** $k = 0, 1, ..., K$ **do**
 6:       $\tilde{A}_{\mathcal{Q}}^{(k)} = DropEdge(A_{\mathcal{Q}});$ ▷ Sample adjacency matrix
 7:       Calculate $\Phi^{(l+1)}(\tilde{A}_{\mathcal{Q}}^{(k)}, A_{\mathcal{T}})$ according to Eq.6
 8:    **end for**
 9:    $S_{subtree}^{(l+1)} = \bigodot_{k=0}^{K} \Phi^{(l+1)}(\tilde{A}_{\mathcal{Q}}^{(k)}, A_{\mathcal{T}});$ ▷ Final subtree assignment matrix
10:    $H_{\mathcal{T}}^{(l+1)} = GNN_{\mathcal{T}}^{(l)}(A_{\mathcal{T}}, concat[H_{\mathcal{T}}^{(l)}, MLP(S^{(l)})]);$
11:    $H_{\mathcal{Q}}^{(l+1)} = GNN_{\mathcal{Q}}^{(l)}(A_{\mathcal{Q}}, concat[H_{\mathcal{Q}}^{(l)}, MLP((S^{(l)})^T)]);$ ▷ GNN update
12:    Compute $S_{gnn}^{(l+1)}$ according to Eq.8 ▷ Final GNN assignment matrix
13:    $S^{(l+1)} = S_{gnn}^{(l+1)} \odot S_{subtree}^{(l+1)};$ ▷ Final assignment matrix
14: **end for**
15: $result = CheckAssign(S^{(L)})$

---

### 4.3 IMPLEMENTATION DETAILS AND COMPLEXITY ANALYSIS

We summarize the overall procedures in Algorithm. 1. The computation of $D^2$Match has two major parts: the subtree and GNN modules. Given $L$ layers and $K$ times of sampling, the complexity of the subtree and GNN modules are $O(L*K*|V_{\mathcal{T}}|*|E_{\mathcal{Q}}|+L*|V_{\mathcal{Q}}|*|E_{\mathcal{T}}|)$ and $O(L*|E_{\mathcal{T}}|+L*|E_{\mathcal{Q}}|+|V_{\mathcal{T}}| * |V_{\mathcal{Q}}|)$, respectively. Since the query graph is often very small, we can treat $|V_{\mathcal{Q}}|$ and $|E_{\mathcal{Q}}|$ as constants. A detailed runtime comparison is in Appendix A.5. Therefore, the overall complexity is reduced to $O(|V_{\mathcal{T}}|+|E_{\mathcal{T}}|)$, attaining linear time complexity. Please refer to Appendix A.1 for more details about the implementation.

## 5 EXPERIMENTS

Here, we conduct extensive experiments to answer the following questions: (1) *How does our proposed $D^2$Match compare to SOTA methods?* (2) *Why GNNs in ours and others yield different results?* (3) *How robust does $D^2$Match perform?* Sec. 5.2-Sec. 5.5 answer the above questions accordingly.

### 5.1 EXPERIMENTAL SETTINGS

**Datasets and Experimental Setup.** We implement our experiments on both synthetic and real-world datasets, which are collected from a large variety of applications. We aim to obtain pairs of query and target graphs, along with labels indicating whether a query is isomorphic to the target. We first generate synthetic data by utilizing ER-random graphs and WS-random graphs (Rex et al., 2020). We keep edge densities the same in both positive and negative samples to ensure consistency in the distribution. This balance avoids potential biases during learning. For the real-world data, we follow the setting in (Rex et al., 2020), including Cox2, Enzymes, Proteins, IMDB-Binary, MUTAG, Aids, and FirstMMDB. We also conduct additional experiments on the Open Graph Benchmark datasets(Hu et al., 2020) and three datasets with continuous features in Appendix A.7. We employ these raw graphs as target graphs and generate the positive query graphs by randomly sampling from the target graphs. The negative query graphs are randomly generated. Similar to the synthetic data, we require the edge density in both positive and negative samples to be as close as possible. We split each dataset into training and testing at a ratio of $4 : 1$ and report the average classification accuracy under the five-fold cross-validation.

Table 1: Overall performance comparison in terms of accuracy.

| | Synthetic | Proteins | Mutag | Enzymes | Aids | IMDB-Binary | Cox2 | FirstMMDB |
|---|---|---|---|---|---|---|---|---|
| SimGNN | $70.5_{\pm 2.72}$ | $96.2_{\pm 0.97}$ | $98.7_{\pm 0.60}$ | $98.6_{\pm 1.08}$ | $96.5_{\pm 0.68}$ | $85.0_{\pm 19.58}$ | $99.9_{\pm 0.22}$ | $82.40_{\pm 0.17}$ |
| NeuroMatch | $65.7_{\pm 8.98}$ | $94.5_{\pm 1.73}$ | $99.2_{\pm 0.22}$ | $97.9_{\pm 1.08}$ | $97.4_{\pm 0.96}$ | $86.5_{\pm 6.51}$ | $100.0_{\pm 0.00}$ | $80.80_{\pm 0.39}$ |
| IsoNet | $50.0_{\pm 0.00}$ | $60.0_{\pm 10.02}$ | $94.1_{\pm 2.54}$ | $91.0_{\pm 7.78}$ | $61.5_{\pm 8.51}$ | $83.1_{\pm 3.69}$ | $95.8_{\pm 3.89}$ | / |
| GMN-embed | $56.6_{\pm 8.61}$ | $93.8_{\pm 2.41}$ | $90.8_{\pm 6.16}$ | $89.4_{\pm 16.44}$ | $78.3_{\pm 6.92}$ | $69.3_{\pm 15.18}$ | $69.7_{\pm 18.20}$ | $69.1_{\pm 30.29}$ |
| GraphSim | $50.0_{\pm 0.00}$ | $82.5_{\pm 0.31}$ | $89.5_{\pm 2.59}$ | $88.2_{\pm 1.79}$ | $75.6_{\pm 6.53}$ | $88.9_{\pm 2.81}$ | $95.5_{\pm 0.94}$ | $86.6_{\pm 9.71}$ |
| GOT-Sim | $53.0_{\pm 2.74}$ | $57.2_{\pm 8.52}$ | $86.8_{\pm 6.92}$ | $68.7_{\pm 14.15}$ | $70.6_{\pm 3.08}$ | $81.3_{\pm 14.60}$ | $94.8_{\pm 1.04}$ | / |
| $D^2$Match | $\mathbf{74.3_{\pm 0.22}}$ | $\mathbf{100.0_{\pm 0.00}}$ | $\mathbf{100.0_{\pm 0.00}}$ | $\mathbf{99.9_{\pm 0.22}}$ | $\mathbf{99.5_{\pm 0.27}}$ | $\mathbf{93.3_{\pm 1.03}}$ | $\mathbf{100.00_{\pm 0.00}}$ | $\mathbf{100.00_{\pm 0.00}}$ |

**Baselines.** To get a fair comparison, we select the following SOTA competitors: SimGNN (Bai et al., 2019), NeuralMatch (Rex et al., 2020), IsoNet (Roy et al., 2022), GMN-embed (Li et al., 2019) GraphSim (Bai et al., 2020), and GOT-Sim (Doan et al., 2021). These all incorporate graph neural networks into subgraph matching. We present the comparison with exact methods in Appendix A.8.

## 5.2 MAIN RESULTS

Table 1 reports the overall performance of all compared models, where each model achieves the best results in all possible settings up to 500 epochs. The accuracy of GOT-Sim and IsoNet on FirstMMDB is omitted due to exceeding time and memories. We observe that

- For the synthetic dataset, the overall performance is much lower than that in the real-world datasets. A reason is that the synthetic dataset is more complicated, e.g., with a higher edge density, than real-world datasets, which makes the matching more challenging. By examining more details, IsoNet, GMN-embed, GraphSim, and GOT-Sim attempt to employ a node-level assignment matrix to capture matching between graphs, which underestimate the importance of global structure. They can only yield around 50% accuracy. SimGNN and NeuroMatch try to learn the global representation and attain the accuracy of 70.5% and 65.7%, respectively.
- For the real-world datasets, $D^2$Match attains superior performance and beats all baselines. Among seven real-world datasets, $D^2$Match attains 100% accuracy in four datasets, i.e., Protein, Mutag, Cox2, and FirstMMDB, while at least 99.5% in Enzymes and Aids, and 93.3% in IMDB-Binary.
- Overall, $D^2$Match has explicitly modeled subtress and consistently attained the best performance among all compared methods. The promising results confirm our theoretical analysis.

## 5.3 BENEFIT OF OUR CORE DESIGN: THE SUBTREE MATCHING

Though GNNs deployed in $D^2$Match and existing learning-based subgraph matching methods, they function differently. $D^2$Match utilizes GNNs to explicitly models subtrees while learning-based methods learn the graph representations via memorizing the data distribution divergence. To validate this, we construct new datasets and denote them with $^*$ by excluding the data distribution effect. We first follow the same way as generating positive samples, then continue to perform edge dropping and insertion on the clipped graphs together to obtain the final negative samples. This strategy aims to make sure the generated samples, both positive and negative, following the same distribution in terms of edges. We also build an new synthetic dataset without following this property, called Synthetic$^+$, for a better comparison. Since SimGNN and NeuoMatch are the best-performing GNN-based methods in Table 1, we select them in the comparison.

Results in Table 2 show that $D^2$Match outperforms SimGNN and NeuoMatch by a much larger margin, achieving $2.5\% - 33.2\%$ improvement. This phenomenon aligns with our hypothesis that the gain of other learning-based methods is distribution-dependent, which results in a significant performance drop on evenly-distributed data. Moreover, the overall performance on Synthetic$^+$ is much better than that on Synthetic. This implies that data following non-even distribution will make the matching much easier. This is in line with the results in Table 1, i.e., the learning-based methods tend to capture the distribution divergence rather than performing matching.

## 5.4 ABLATION STUDIES

**Effect of $L$, the depth of a subtree.** We test the effect of the depth of a subtree, i.e., the number of the hidden layers, and change it from 1 to 7. Results in Fig. 2(a) shows that $D^2$Match reaches

Table 2: Results of experimenting the uniformly distributed data in terms of accuracy.

| | Synthetic$^+$ | Synthetic | Proteins$^*$ | Mutag$^*$ | Enzymes$^*$ | Aids$^*$ | IMDB-Binary$^*$ | Cox2$^*$ | FirstMMDB$^*$ |
|---|---|---|---|---|---|---|---|---|---|
| SimGNN | $84.1_{\pm 3.40}$ | $70.5_{\pm 2.72}$ | $64.6_{\pm 3.36}$ | $80.3_{\pm 6.18}$ | $76.0_{\pm 2.12}$ | $73.2_{\pm 6.06}$ | $72.0_{\pm 2.45}$ | $88.2_{\pm 2.05}$ | $53.2_{\pm 6.61}$ |
| NeuroMatch | $74.5_{\pm 2.57}$ | $65.7_{\pm 8.98}$ | $52.8_{\pm 4.76}$ | $90.4_{\pm 2.88}$ | $86.6_{\pm 3.64}$ | $75.6_{\pm 17.78}$ | $60.4_{\pm 10.11}$ | $91.0_{\pm 5.70}$ | $50.0_{\pm 0.00}$ |
| $D^2$Match | $\mathbf{86.6}_{\pm 1.44}$ | $\mathbf{74.3}_{\pm 0.22}$ | $\mathbf{83.4}_{\pm 2.97}$ | $\mathbf{99.2}_{\pm 0.84}$ | $\mathbf{96.0}_{\pm 2.16}$ | $\mathbf{95.0}_{\pm 1.41}$ | $\mathbf{90.2}_{\pm 1.79}$ | $\mathbf{99.8}_{\pm 0.45}$ | $\mathbf{86.4}_{\pm 7.44}$ |

its best performance when the number of layers is 6. $D^2$Match only needs a few layers to achieve a decent performance, suggesting it can scale up to large size graphs.

**Effect of $K$, the times of samples.** Intuitively, sampling more data to train the model will yield better performance. We vary $K$ from 1 to 7 and show the results in Fig. 2(b). Surprisingly, the results show that by sampling five times, we can obtain the best performance on all datasets. This demonstrates that $D^2$Match can attain decent performance in a low computation cost.

We also ablate $D^2$Match with circles and node attributes at different settings, and all experiments show results consistent with our theoretical analysis. Please refer to Appendix A.3 for more details.

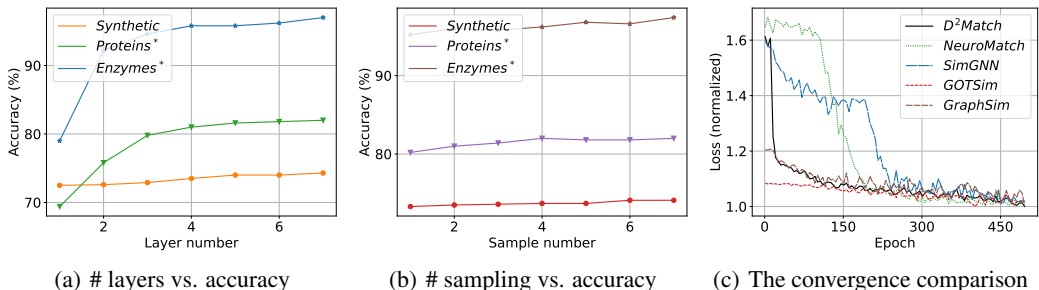

(a) # layers vs. accuracy  (b) # sampling vs. accuracy  (c) The convergence comparison

Figure 2: We conduct sensitivity analysis on our $D^2$Match by varying the number of layers and sampling. In Fig. 2(c), we present the convergence curve on our $D^2$Match and four strong baselines.

## 5.5 Convergence Analysis

Figure 2(c) provides the training loss of $D^2$Match and four baselines on Synthetic, where we only select baselines with the same loss functions as ours, such as MSE or CE, for a fair comparison. The results show that (1) $D^2$Match converges the fastest due to its power of explicitly modeling the subtrees. (2) NeuroMatch and SimGNN perform matching through learning graph-level representations, which need more epochs to converge for capturing the local structure. (3) GOTSim and GraphSim attain the lowest loss in the beginning but show the weakest convergence ability compared to others because they can only capture the node-level representations and fail to learn meaningful subgraph matching. Consequently, they yield the worst performance as reported in Table 1.

## 6 Conclusion and Future Work

In this paper, we propose $D^2$Match for subgraph matching, which degenerates the subgraph matching problem into perfect matching in a bipartite graph and prove that the matching procedure can be implemented via the built-in tree-structure aggregation on GNNs, which yields polynomial time complexity. We also incorporate circle structures and node attributes to boost the matching. Finally, we conduct extensive experiments to show that $D^2$Match achieves significant improvement over competitive baselines and indeed exploits subtrees for the matching, which is different from existing learning-based methods for memorizing the data distribution divergence.

$D^2$Match can be further explored in several promising directions. First, we can investigate more degeneracy mechanisms to tackle more complicated graphs. Second, we can extend our $D^2$Match to more real-world applications, e.g., document matching, to know its capacity.

## REPRODUCIBILITY STATEMENT

The supplemental material includes the code for our experiments. An detailed description of the datasets used in the experiments is provided in Appendix A.6.

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

## A   APPENDIX

**Theorem A.1.** Given a target graph $G_{\mathcal{T}}(V_{\mathcal{T}}, E_{\mathcal{T}})$ and a query graph $G_{\mathcal{Q}}(V_{\mathcal{Q}}, E_{\mathcal{Q}})$, if $G_{\mathcal{Q}} \subset G_{\mathcal{T}}$, and the subtree generation function $\Psi$ as defined in Eq. (2) meets the following condition:

$$\forall \text{ graph pair } (G_{\mathcal{S}}, G), \text{ if } G_{\mathcal{S}} \subset G, \text{ then } \Psi(G_{\mathcal{S}}) \subset \Psi(G), \tag{9}$$

then there exists an injective function $f : V_{\mathcal{Q}} \to V_{\mathcal{T}}$, ensuring the $l$-hop subtrees of the subgraph is isomorphic to the subtrees of the corresponding subgraph:

$$\forall l \geq 1, q \in V_{\mathcal{Q}}, t = f(q) \in V_{\mathcal{T}} \Rightarrow T_q^{(l)} \subset T_t^{(l)}, \tag{10}$$

*Proof.* According to the definition of subgraph matching (McCreesh et al., 2018), when $G_{\mathcal{Q}}$ is a subgraph of $G_{\mathcal{T}}$, there must exists an injective function $f : V_{\mathcal{Q}} \to V_{\mathcal{T}}$, such that $\forall q_i, q_j \in V_{\mathcal{Q}}, (q_i, q_j) \in E_{\mathcal{Q}} \Rightarrow (f(q_i), f(q_j)) \in E_{\mathcal{T}}$. For any subgraph in the query graph, e.g., $S(V_S, E_S) \in G_{\mathcal{Q}}$, we always have a subgraph in the original graph $G_{\mathcal{T}}$, denoted as $G_S(V_G, E_G)$, that corresponds to the set of the query node as $V_G = f(V_S)$. This tells us that $S \subset G_S$. According to this, consider any given node from $V_{\mathcal{Q}}$: $q \in V_{\mathcal{Q}}$, $S_q^{(l)}$ is a subgraph of $G_{\mathcal{Q}}$ and its image $G_{S_q^{(l)}}$ in $G_{\mathcal{T}}$, i.e. $S_q^{(l)} \subset G_{S_q^{(l)}}$. By definition, the node in $S_q^{(l)}$ or $G_{S_q^{(l)}}$ is at most $l$-hop from node q or $t = f(q)$, we know that $G_{S_q^{(l)}}$ must be a subgraph of $S_t^{(l)}$, i.e., $G_{S_q^{(l)}} \subset S_t^{(l)}$. Put all together, we have $S_q^{(l)} \subset G_{S_q^{(l)}} \subset S_t^{(l)}$. Based on the listed constrain, we then have $T_q^{(l)} \subset T_t^{(l)}$.

$\square$

**Theorem A.2.** Given a node q in the query graph and a node t in the target graph, the following three conditions are equivalent:

1) $T_q^{(l+1)} \subset T_t^{(l+1)}$.
2) There exists an injective function on the neighborhood of these nodes as $f : N(q) \to N(t)$, s.t. $\forall q_i \in N(q), t_i = f(q_i), T_{q_i}^{(l)} \subset T_{t_i}^{(l)}$.
3) There exists a perfect matching on the bipartite graph $B^{(l)}(N(t), N(q), E)$, where $\forall t_j \in N(t), q_i \in N(q), (t_j, q_i) \in E$ if and only if $T_{q_i}^{(l)} \subset T_{t_j}^{(l)}$.

We prove this theorem by introducing the following two theorem. Theorem A.3 shows that condition 1) is equivalent to condition 2), i.e. the WL subtree isomorphism test can be accomplished in a recursive manner then prove Theorem. A.4 that the condition 2) equals to condition 3) which means every iteration in the recursive process equals to examine the existence of a perfect matching, respectively.

**Theorem A.3.** Given a node q in the query graph and a node t in the target graph, the following two conditions are equal:
1) $T_q^{(l+1)} \subset T_t^{(l+1)}$, where $l$ is an integer and $l \geq 1$.
2) There exists an injective function on the neighboring set of these nodes as $f : N(q) \to N(t)$, s.t.$\forall q_i \in N(q), t_i = f(q_i), T_{q_i}^{(l)} \subset T_{t_i}^{(l)}$.

*Proof.* We assume $f_q$ is a subtree isomorphism injective function in the condition 1), that $\forall$ node $u, v \in T_q^{(l+1)}, (u, v)$ is an edge of $T_q^{(l+1)} \Rightarrow ((f_q(u), f_q(v))$ is an edge of $T_t^{(l+1)}$. Similarly We also assume $f_{q_i}$ is subtree isomorphism injective in the condition 2).
On the one hand, if condition 1) is true then $f_q$ exists. Using the property of WL tree, we have $\forall q_i \in N(q), T_{q_i}^{(l)} \subset T_q^{(l+1)}$, which means the $l$-order WL tree of any node $q_i$ in q's neighbourhood belongs to the $l + 1$-order WL tree originate from the node q. This suggests that $f_q$ maps $T_{q_i}^{(l)}$ into a tree $T_{f(q_i)}^{(l)} = T_{t_i}^{(l)}$, which is a subtree of $T_t^{(l+1)}$. Then the condition 2) is true.

On the other hand, if condition 2) holds, then we define the mapping as $f_q(v) = \begin{cases} f_{q_i}(v), & v \in T_{q_i}^{(l)} \\ q, & v = q \end{cases}$.

Here, the function $f_b(v)$ is a standard injective function $T_{b_i}^{(l)}$ This implies this is a subtree isomorphic mapping, so 1) holds.

$\square$

The above theorem provides a recursive solution to the WL subtree isomorphism algorithm. Intuitively, we can maintain an indicator matrix $S^{(l)} \in R^{|V_\mathcal{T}| \times |V_\mathcal{Q}|}$, where $S_{tq}^{(l)} = \begin{cases} 1, & T_q^{(l)} \subset T_t^{(l)} \\ 0, & \text{else} \end{cases}$.
This matrix captures the relation between all pairs of nodes and thus can be used for recursion update. Next, we will show that the update process can be implemented as a perfect matching problem, i.e., what makes condition 2) true is equivalent to finding a perfect matching on a bipartite graph, as shown in the following theorem:

**Theorem A.4.** Assume the neighboring set of node t and q as $X = N(t)$ and $Y = N(q)$, respectively. Accordingly, we form a bipartite graph as $B_{t,q}^{(l)}(X, Y, E)$. Here, we define the edges as $E = \{(t_i, q_j) : T_{q_i}^{(l)} \subset T_{t_j}^{(l)}\}$, where $t_i$ and $q_j$ represent the $i$th and $j$th neighbour of node t and q, respectively. Under this setting, the injective function $f$ from the condition 2) in Theorem. A.3 induces a perfect matching.

*Proof.* The injective function $f$ of condition 2) in Theorem A.3 maps every node $q_i$ in $N(q)$ to $t_i = f(q_i) \in N(t)$ and $T_{q_i}^{(l)} \subset T_{t_i}^{(l)}$ holds. While $T_{q_i}^{(l)} \subset T_{t_i}^{(l)}$ means $(q_i, t_i) \in E$, the injective $f$ naturally corresponds every node $q_i$ to an edge $(q_i, t_j)$. Since $f$ is an injective function, $q_{i_1} \neq q_{i_2} \Rightarrow t_{i_1} \neq t_{i_2}$, indicating that all these edges $(q_i, t_i), i = 1, ..., |N(q)|$ are different, which actually forms a perfect matching. $\square$

**Theorem A.5.** Given the sampled query graph and the target graph, we can construct their adjacency matrices , $\tilde{A}_\mathcal{Q}$ and $A_\mathcal{T}$, and the degree matrix of the sampled query graph $\tilde{D}_\mathcal{Q} = \text{diag}(\sum_s ((\tilde{A}_\mathcal{Q})_{:s}))$. Here, we denote the indicator matrix at the $l$-th hop as $S^{(l)}$. To check the validity of $|N(W)| \geq |W|$, we can check whether each element of $\Phi$ is true or not, where $\Phi := Z_{N(W)} \geq 1$, $Z_{N(W)} = \text{aggregate}_{\text{sum}}(A_\mathcal{T}, Z_W^T)$ and $Z_W = \text{aggregate}_{\text{max}}(\tilde{D}_\mathcal{Q}^{-1} \cdot \tilde{A}_\mathcal{Q}, (S^{(l)})^T)$.

*Proof.* For each node pair t, q and their corresponding $W = N'(q)$ in the sampled query graph, We first transform the neighboring set of $W$, i.e., $N(W)$, as following:

$$\begin{aligned} N(W) &= \{t_i \in N(t) | \exists q_j \in W = N'(q), \text{s.t.} T_{q_j}^{(l)} \subset T_{t_i}^{(l)}\} \\ &= \{t_i \in N(t) | \exists q_j \in W = N'(q), \text{s.t.} S_{t_i, q_j} = 1\} \\ &= \{t_i \in N(t) | \max_{q' \in N'(q)} S_{t_i, q'}^{(l)} = 1\} \\ &= N(t) \cap \{t_i | \max_{q' \in N'(q)} S_{t_i, q'}^{(l)} = 1\} \\ &= N(t) \cap M(q) \end{aligned}$$

(11)

Let $M(q) = \{t_i | \max_{q' \in N'(q)} S_{t_i, q'}^{(l)} = 1\}$, we can compute $M(q)$ via a standard maximizing aggregation process on the sampled adjacency matrix $\tilde{A}_\mathcal{Q}$, in which treats the indicator matrix $(S^{(l)})^T \in R^{|V_\mathcal{Q}| \times |V_\mathcal{T}|}$ as node attributes. This process will output the representation of node q as follows,

$$z_{q,:} = max\{(S^{(l)})_{j,:}^T, \forall j \in N'(q)\},$$

(12)

The obtained vector $z_{q,:}$ is to represent $M(q)$ where $z_{qi} = \begin{cases} 1, & i \in M(q) \\ 0, & \text{else} \end{cases}$. We rewrite this into a matrix format as

$$Z_W = \text{aggregate}_{\text{max}}(\tilde{A}_\mathcal{Q}, (S^{(l)})^T)$$

(13)

where $Z_W \in R^{|V_\mathcal{Q}| \times |V_\mathcal{T}|}$ and its q-th row vector is $z_{q,:}$.

Recall that we demand $N(W) = N(\text{t}) \cap M(\text{q})$. After acquiring $M(\text{q})$, we can compute the $|N(W)|$ as follows,

$$\begin{cases} |M(\text{q})| = \sum_i z_{\text{q},i} \\ |N(W)| = \sum_i z_{\text{q},i}, i \in N(\text{t}) \end{cases} \tag{14}$$

In essence, this is to implement a summation aggregation on the target graph using the node representation $Z_W$, i.e.,

$$Z_{N(W)} = \text{aggregate}_{\text{sum}}(A_\mathcal{T}, Z_W^T) \tag{15}$$

where $Z_{N(W)} \in R^{|V_\mathcal{T}| \times |V_\mathcal{Q}|}$ is an integer matrix and its element $(\text{t}, \text{q})$ shows the score of $|N(W)|$ between node t and q. This transformation converts the counting operation as aggregation such that we can check the aggregated values to determine whether there is a perfect matching. Given a node pair $(\text{t}, \text{q})$, we have $|N(W)| = [Z_{N(W)}]_{\text{tq}}$ and $|W| = |N'(\text{q})| = \sum_s [\tilde{A}_\mathcal{Q}]_{\text{q}s}$. Therefore, the question becomes to check whether $[Z_{N(W)}]_{\text{tq}} \geq \sum_s [\tilde{A}_\mathcal{Q}]_{\text{q}s}$ holds. We can then derive the perfect matching as follows:

$$[Z_{N(W)}]_{\text{tq}} \geq \sum_s [\tilde{A}_\mathcal{Q}]_{\text{q}s}$$

$$\Leftrightarrow [Z_{N(W)}]_{\text{tq}} / \sum_s [\tilde{A}_\mathcal{Q}]_{\text{q}s} \geq 1$$

$$\Leftrightarrow [\text{aggregate}_{\text{sum}}(A_\mathcal{T}, Z_W^T)]_{\text{tq}} / \tilde{d}_\text{q} \geq 1 \tag{16}$$

$$\Leftrightarrow [\text{aggregate}_{\text{sum}}(A_\mathcal{T}, Z_W^T) \cdot \tilde{D}_\mathcal{Q}^{-1}]_{\text{tq}} \geq 1$$

$$\Leftrightarrow [\text{aggregate}_{\text{sum}}(A_\mathcal{T}, Z_W^T \cdot \tilde{D}_\mathcal{Q}^{-1})]_{\text{tq}} \geq 1$$

where $\tilde{d}_\text{q}$ is the degree of node q in the sampled graph. The degree matrix of the sample graph is defined as $\tilde{D}_\mathcal{Q} = \text{diag}[\sum_s ((\tilde{A}_\mathcal{Q})_{:s})]$. Now recall that $\Phi$ is the matrix whose $(\text{t}, \text{q})$ element is the comparison result of $|N(W)|$ and $|W|$ of $(\text{t}, \text{q})$, according to eq 16, we have:

$$\Phi^{(l+1)}(\tilde{A}_\mathcal{Q}, A_\mathcal{T}) = \text{aggregate}_{\text{sum}}(A_\mathcal{T}, Z_W^T \cdot \tilde{D}_\mathcal{Q}^{-1}) \geq 1, \tag{17}$$

where

$$\begin{aligned} Z_W^T \cdot \tilde{D}_\mathcal{Q}^{-1} &= [\text{aggregate}_{\text{max}}(\tilde{A}_\mathcal{Q}, (S^{(l)}))]^T \cdot \tilde{D}_\mathcal{Q}^{-1} \\ &= [\tilde{D}_\mathcal{Q}^{-1} \cdot \text{aggregate}_{\text{max}}(\tilde{A}_\mathcal{Q}, (S^{(l)}))]^T \\ &= [\text{aggregate}_{\text{max}}(\tilde{D}_\mathcal{Q}^{-1} \cdot \tilde{A}_\mathcal{Q}, (S^{(l)}))]^T \end{aligned} \tag{18}$$

$\square$

**Theorem A.6.** Every chordless cycle is atomic. Every chordless cycle $\mathcal{C}_\mathcal{Q}$ in an induced subgraph $G_\mathcal{Q}$ must correspond to a chordless cycle $\mathcal{C}_\mathcal{T}$ in the origin graph $G_\mathcal{T}$.

*Proof.* Chordless cycle does not have any chord, thus there is no smaller cycle in the chordless cycle, which means chordless cycle is atomic. Assuming $G_\mathcal{Q}$ is a subgraph of $G_\mathcal{T}$, every node of $\mathcal{C}_\mathcal{Q}$ must correspond to a node in $G_\mathcal{T}$, and these nodes form a circle $\mathcal{C}_\mathcal{T}$ in $G_\mathcal{T}$. Since $G_\mathcal{Q}$ is an induced subgraph of $G_\mathcal{T}$, if $\mathcal{C}_\mathcal{T}$ has a chord, then $\mathcal{C}_\mathcal{Q}$ must have a chord, which contradicts the condition that $\mathcal{C}_\mathcal{Q}$ is a chordless graph. $\square$

## A.1 IMPLEMENTATION DETAILS

The python implementation of $D^2$Match is available at:

```
https://www.dropbox.com/sh/8pvj8drvj0l2zou/AAB5j7e7frVwMiun1QcCNbMFa?
dl=0
```

At the beginning of subtree isomorphism test, the model needs an initial indicator matrix $S_{subtree}^{(0)}$ as the input of the first iteration. According to the definition of the indicator matrix, $S_{subtree}^{(0)}$ shows the isomorphism relation between the subtree of 0-hop neighbors, which are the nodes themselves in this case. Since all nodes will be isomorphic to each other if not considering the node attributes, the indicator matrix $S_{subtree}^{(0)}$ is actually a similarity matrix w.r.t node attributes. To get a similarity matrix of attributes, we can either directly calculate the similarity between nodes or employ neural

networks on these attributes to learn the matrix. In our model, we implement both methods to initialize the matrix, called the initialization of the raw and the learnable:

$$Raw : S^{(0)}_{subtree} = CosineSimilarity(X_\mathcal{T}, X_\mathcal{Q}) = Norm(X_\mathcal{T}) \cdot Norm(X_\mathcal{Q}^T)$$

$$Learnable : S^{(0)}_{subtree} = MLP(X_\mathcal{T}) \cdot MLP(X_\mathcal{Q})^T \tag{19}$$

where the raw initialization is to calculate the cosine similarity between the nodes' attributes, and the learnable initialization employs a MLP to generate hidden representations of nodes and compute their dot similarities.

In practice, we find the raw initialization performs better. This is because the node attributes of datasets are usually binary categorical vectors, which induces clear identification information of the nodes and can be easily captured by cosine similarity.

Our implementation of the GNN block in the model is slightly different from the description. Specifically, we use compute the similarity of each pair of nodes as:

$$[S^{(l+1)}_{gnn}]_{ij} = MLP(concat([H^{(l)}_\mathcal{T}]_i, [H^{(l)}_\mathcal{Q}]_j)). \tag{20}$$

The main difference is that we do not output a $|V_\mathcal{T}| \times |V_\mathcal{Q}|$ matrix, but a $|V_\mathcal{T}| \times |V_\mathcal{Q}| \times |D^{(l+1)}|$ tensor, where $D^{(l+1)}$ denotes the hidden dim of $l+1$ layer. The intuition is that a tensor that represents the node pairs' similarity with vectors can retain more information than a similarity matrix with scalar elements. In this setting, the final indicator matrix $S^{(l+1)}$ can not be generated as $S^{(l+1)} = S^{(l+1)}_{gnn} \odot S^{(l+1)}_{subtree}$, because $S^{(l+1)}_{subtree} \in R^{|V_\mathcal{T}| \times |V_\mathcal{Q}|}$ but $S^{(l+1)}_{gnn} \in R^{|V_\mathcal{T}| \times |V_\mathcal{Q}| \times |D^{(l+1)}|}$. Thus we broadcast $S^{(l+1)}_{subtree}$ to $\tilde{S}^{(l+1)}_{subtree}$ where $\forall k \in [0, D^{(l+1)}), [\tilde{S}^{(l+1)}_{subtree}]_{ijk} = [S^{(l+1)}_{subtree}]_{ij}$ and the final indicator matrix $S^{(l+1)} = S^{(l+1)}_{gnn} \odot \tilde{S}^{(l+1)}_{subtree}$

At the end of our models, we get the subtree indicator matrix $S^{(L)}_{subtree}$ and the GNN indicator matrix $S^{(L)}_{gnn}$. The model will output the final score from $S^{(L)}_{subtree}$ and $S^{(L)}_{gnn}$, respectively. For the subtree module, we check whether the indicator matrix is feasible to induce the subgraph isomorphism. Note that for a node $i$ in the target graph and a node $j$ in the query graph, $i$ is possible to match $j$ unless $[S^{(L)}_{subtree}]_{ij} = 1$. So we check whether the subtree indicator matrix meets the following two conditions:

1) Every node in a query graph should match at least one node in the target graph:

$$\forall j, \max_i (S^{(L)}_{subtree})_{ij} = 1$$

$$\Leftrightarrow \sum_j \max_i (S^{(L)}_{subtree})_{ij} = |V_\mathcal{Q}| \tag{21}$$

$$\Leftrightarrow \sum_j \max_i (S^{(L)}_{subtree})_{ij} / |V_\mathcal{Q}| = 1$$

2) The number of nodes in the target graph that match at least one node in the query graph is more than the number of nodes of query graph:

$$\sum_i \max_j (S^{(L)}_{subtree})_{ij} \geq |V_\mathcal{Q}| \Leftrightarrow \sum_i \max_j (S^{(L)}_{subtree})_{ij} / |V_\mathcal{Q}| \geq 1 \tag{22}$$

To make the subtree model differentiable, we use a learnable sigmoid to replace all the logical judgment in the model:

$$LSigmoid(x) = \sigma(ax + b) \tag{23}$$

where $a, b$ are learnable parameters; $\sigma$ is the sigmoid function. The result of subtree module can be fomulated as:

$$r_{subtree} = LSigmoid(\sum_i \max_j (S^{(L)}_{subtree})_{ij} / |V_\mathcal{Q}|) \cdot LSigmoid(\sum_j \max_i (S^{(L)}_{subtree})_{ij} / |V_\mathcal{Q}|) \tag{24}$$

For the GNN module, we employ the neural tensor network(NTN) (Bai et al., 2019) and generate a score according to the output of NTN and the aggregated indicator tensor:

$$r_{gnn} = \sigma(MLP(concat[NTN(H^{(L)}_\mathcal{T}, H^{(L)}_\mathcal{Q}), \sum_i \sum_j S^{(L)}_{subtree}])) \tag{25}$$

Where $H_{\mathcal{T}}^{(L)}, H_{\mathcal{Q}}^{(L)}$ are the node representations generated by the GNNs. $NTN$ is the NTN layer.

The final prediction is:

$$r = r_{gnn} \cdot r_{subtree} \tag{26}$$

Although the model's prediction is obtained by integrating the two modules, we can not directly train the model through the final score $r$ because it will bring difficulties in the training process. When fitting a negative sample, the resulting subtree module tends to be zero, forcing the overall gradient to be zero which hinders the training of the GNN module.

Therefore, we train the two blocks with different objectives. For the subtree module which aims to learn the isomorphism relation, the result should be either 0 for not matching or 1 for matching. So we employ MAE loss to enforce the results to be either 0 or 1. For the GNN module, we use MSE to encourage the output of GNNs to capture the similarity. Suppose the ground-truth label is $y$, and our loss function is

$$L = MSE(r_{gnn}, y) + MAE(r_{subtree}, y) \tag{27}$$

Both our model and all baselines use the Adam as optimizer and set the learning rate to $3e-4$. To ensure fairness, we set all models with adjustable number of layers to 5 layers, and set the hidden dimension to 10.

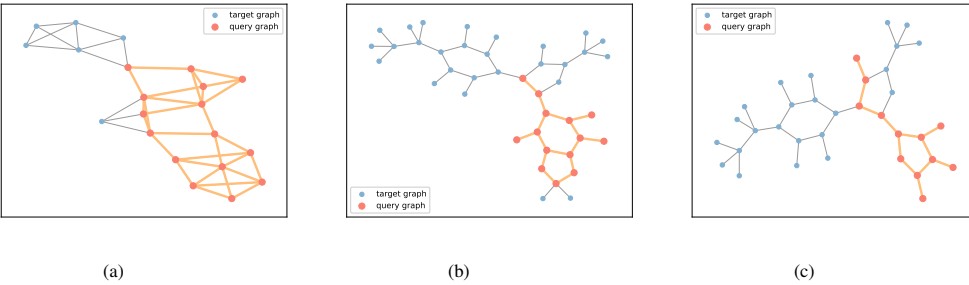

(a)          (b)          (c)

Figure 3: The detected subgraphs by $D^2$Match

## A.2 $D^2$MATCH AT WORK

Recall that $D^2$Match learns an indicator matrix to capture pairwise similarities. It plays the role of permutation matrix in matching, allowing us to pinpoint the matched subgraph. This is particularly useful since the exact position is required for some downstream applications such as web search. In comparison, other learning-based methods are unable to pinpoint local correspondences, but only establish the existence of a matching. We provide a visualization of the matched subgraph to better understand the problem difficulty and the effectiveness of our method, as shown in the Fig. 3.

## A.3 ABLATION STUDIES

We perform ablation studies for the GNN module, subtree module, and chordless cycle.

The GNN module in our analysis will capture the distributional features on the graph, such as the edge density difference between classes. The GNN module is thus essential for datasets with multiple distributions, also called biased data. We run experiments on both the biased and unbiased synthetic datasets to show the performance of our method and its variation that without the GNN module, as shown in Table 3. $D^2$Match outperforms $D^2$Match without the GNN module as our theory predicts. But our GNN module shares the same weaknesses as the other GNN models when dealing with evenly distributed data. We observe that the performance of the GNN module drops significantly on hard datasets similar to other GNN models. The subtree module can significantly improve the performance because it harnesses the property of subgraph-matched data, making it robust to data's distribution. Our subtree module outperformed the GNN module on all datasets in our ablation study, demonstrating its effectiveness.

Table 3: The ablation study of D2Match module

| | Synthetic | Synthetic$^+$ | Proteins | Proteins$^*$ | IMDB-Binary | IMDB-Binary$^*$ | FirstMMDB | FirstMMDB$^*$ |
|---|---|---|---|---|---|---|---|---|
| $D^2$Match (gnn only) | $61.1_{\pm13.31}$ | $70.2_{\pm18.58}$ | $95.2_{\pm1.04}$ | $77.2_{\pm8.11}$ | $50.0_{\pm0.00}$ | $64.4_{\pm19.73}$ | $69.7_{\pm26.98}$ | $67.8_{\pm24.38}$ |
| $D^2$Match (subtree only) | $70.0_{\pm2.09}$ | $74.8_{\pm2.56}$ | $100.0_{\pm0.00}$ | $82.0_{\pm2.92}$ | $92.9_{\pm1.04}$ | $82.8_{\pm4.02}$ | $100.0_{\pm0.0}$ | $72.0_{\pm6.20}$ |
| $D^2$Match | $72.7_{\pm4.45}$ | $86.6_{\pm1.44}$ | $100.0_{\pm0.00}$ | $83.4_{\pm2.97}$ | $93.3_{\pm1.03}$ | $90.2_{\pm1.79}$ | $100.0_{\pm0.0}$ | $86.4_{\pm7.44}$ |

Table 7: The hard dataset details

| | Synthetic$^+$ | Proteins$^*$ | Mutag$^*$ | Enzymes$^*$ | Aids$^*$ | IMDB-Binary$^*$ | Cox2$^*$ | FirstMMDB$^*$ |
|---|---|---|---|---|---|---|---|---|
| Average nodes (target) | 40.0 | 38.8 | 18.2 | 31.5 | 14.7 | 19.0 | 41.3 | 1376.7 |
| Average nodes (query) | 15.0 | 11.4 | 9.1 | 15.4 | 4.4 | 14.2 | 15.0 | 15.0 |
| Average edges (target) | 259.5 | 146.7 | 40.2 | 120.6 | 30.0 | 177.1 | 87.0 | 6141.6 |
| Average edges (query) | 67.3 | 35.5 | 17.6 | 52.6 | 7.1 | 102.6 | 29.9 | 45.6 |

We also perform the ablation study on the Synthetic dataset to test the effect of chordless cycles, as shown in Table 4. Results show the chordless cycles boost the performance with limited extra time consumption.

| | Synthetic | RunTime |
|---|---|---|
| $D^2$Match | $74.3_{\pm1.60}$ | 19.7s/epoch |
| $D^2$Match (w/o cc) | $72.7_{\pm4.45}$ | 10.3s/epoch |

Table 4: The ablation study of cc

| | proteins | mutag |
|---|---|---|
| Seed(0) | $100.0_{\pm0.00}$ | $100.0_{\pm0.00}$ |
| Seed(1) | $100.0_{\pm0.00}$ | $100.0_{\pm0.00}$ |
| Seed(2) | $100.0_{\pm0.00}$ | $100.0_{\pm0.00}$ |
| Fixed | $100.0_{\pm0.00}$ | $100.0_{\pm0.00}$ |

Table 5: Random seed comparison

| | Training(s/epoch) | Inference(s/epoch) |
|---|---|---|
| SimGNN | 1.732 | 0.385 |
| NeuroMatch | 2.234 | 0.311 |
| GMN-embed | 1.850 | 0.290 |
| GraphSim | 3.223 | 0.433 |
| IsoNet | 10.553 | 1.939 |
| $D^2$Match-Subtree(S=2) | 2.940 | 0.456 |
| $D^2$Match-Subtree(S=3) | 3.889 | 0.581 |
| $D^2$Match-Subtree(S=4) | 4.410 | 0.673 |
| $D^2$Match-Subtree(S=5) | 5.143 | 0.750 |
| $D^2$Match-GNN | 2.678 | 0.495 |
| $D^2$Match | 8.163 | 1.114 |

Table 6: Runtime analysis

## A.4 RANDOM EFFECT

Although our experiments do not rely on random seeds, a random split may affect the results. To test this, we set up several random seeds and permute the raw data order before getting the five-fold. We experiment on the Protein and Mutag datasets with trivial random seed 0,1,2 and obtain nearly identical performance. See Table 5.

While other methods based on GNNs tend to capture the divergence of distributions in the training set and hence are easily affected by randomness, our subtree module performs the matching explicitly by the degeneracy property, as opposed to modeling the data distribution in others, hence ours is insensitive to data partitioning.

## A.5 RUNTIME ANALYSIS

We add the runtime analysis experiment as follows. We compare our method with baselines on the synthetic dataset and record the training and inference time (second) per epoch. The results are shown in Table 6.

Our model is slower than some strong baselines like SimGNN and NeuroMatch in the experiment because they deal with the graph-level representations. Our model is faster than IsoNet, which performs edge-level matching.

We conduct an additional ablation study to explore the time consumption of each module in our model. The results show that the time consumption of our model mainly comes from the sampling in the subtree module whose running time is linearly related to the sampling number. When we set

Table 8: The dataset details

|  | Synthetic | Proteins | Mutag | Enzymes | Aids | IMDB-Binary | Cox2 | FirstMMDB |
|---|---|---|---|---|---|---|---|---|
| Average nodes (target) | 40.0 | 39.1 | 17.9 | 33.0 | 15.7 | 19.8 | 41.3 | 1376.5 |
| Average nodes (query) | 15.0 | 14.4 | 9.0 | 14.8 | 7.9 | 14.6 | 14.4 | 15.0 |
| Average edges (target) | 241.7 | 146.5 | 39.5 | 125.6 | 32.4 | 193.1 | 87.0 | 6144.3 |
| Average edges (query) | 50.6 | 68.9 | 25.1 | 75.3 | 17.1 | 141.0 | 42.8 | 68.1 |

the sampling number as 2, the running time is on par with the others. Furthermore, the running time for the GNN module is the same as for the other baselines. In sum, we observe that our model's scalability is acceptable as both complexity analysis and empirical running time show ours is slower than others only by a constant factor.

## A.6 DATASET DETAILS

We describe the average node number and average edge number of the target graph and query graph in the Table 8 and Table 7. Except the hard datasets, we generate 1000 graph pairs for Synthetic, Proteins, Mutag, Enzymes, Cox2 and FirstMMDB and 2000 graph pairs for Aids and IMDB-Binary which have smaller graph size. For the hard dataset, we uniformly generate 500 graph pairs.

## A.7 RESULTS ON MORE DATASETS

We conduct experiments on the OGB benchmark dataset (Hu et al., 2020), including Ogbg-molhiv and Ogbg-molpcb. We follow the same strategy in the paper to construct normal and hard versions for these datasets and choose the best-performing baselines for comparison, including SimGNN and NeuroMatch. We present new results in Table 9.

We find that our model performs slightly better than others on normal datasets while gaining a significant advantage over baselines on hard datasets. These results are consistent with our previous experiments, demonstrating that our model exploits the subgraph matching property, rather than simply modeling the divergence of the data distribution as other GNNs.

We experiment on continuous features from the MNIST, CIFAR10 and PPI datasets, as these are constructed from vision data(Dwivedi et al., 2020) or biological information data(Zitnik & Leskovec, 2017). We As expected, our model achieves consistent performance as well. See Table 10.

|  | ogb-molhiv | ogb-molhiv* | ogb-molpcba | ogb-molpcba* |
|---|---|---|---|---|
| SimGNN | $99.4_{\pm 0.65}$ | $81.6_{\pm 2.70}$ | $99.8_{\pm 0.27}$ | $86.2_{\pm 2.28}$ |
| NeuroMatch | $98.3_{\pm 1.68}$ | $86.0_{\pm 3.54}$ | $99.8_{\pm 0.27}$ | $90.6_{\pm 3.51}$ |
| $D^2$Match | $99.8_{\pm 0.27}$ | $99.6_{\pm 0.54}$ | $100.0_{\pm 0.00}$ | $100.0_{\pm 0.00}$ |

Table 9: Obg dataset performance comparison

|  | Cifar10 | MNIST | PPI |
|---|---|---|---|
| SimGNN | $89.0_{\pm 21.82}$ | $98.5_{\pm 0.93}$ | $77.0_{\pm 24.67}$ |
| NeuroMatch | $98.1_{\pm 1.14}$ | $95.9_{\pm 1.34}$ | $50.0_{\pm 0.00}$ |
| $D^2$Match | $99.3_{\pm 0.27}$ | $98.8_{\pm 1.15}$ | $98.8_{\pm 1.06}$ |

Table 10: Continues dataset performance

## A.8 COMPARISON WITH EXACT METHOD

we compare exact matching solutions, including VF2[1] and ISMAGS[2]. By nature, we know that exact matching methods obtain 100 % accuracy.

As a trade-off between accuracy and execution time, we make the comparison inspired by the setup in NeuroMatch (Rex et al., 2020). We say an execution succeeds when its run time is less than 60s. We compare the success rate of the exact methods by varying the query graph size from 10 to 50 on the synthetic data, as shown in Figure.4.

We show in our experiment that the failure of exact matching methods increases significantly when the target graph has more than 30 nodes, compared to our stable performance, indicating the incompetence of these methods on large-scale datasets.

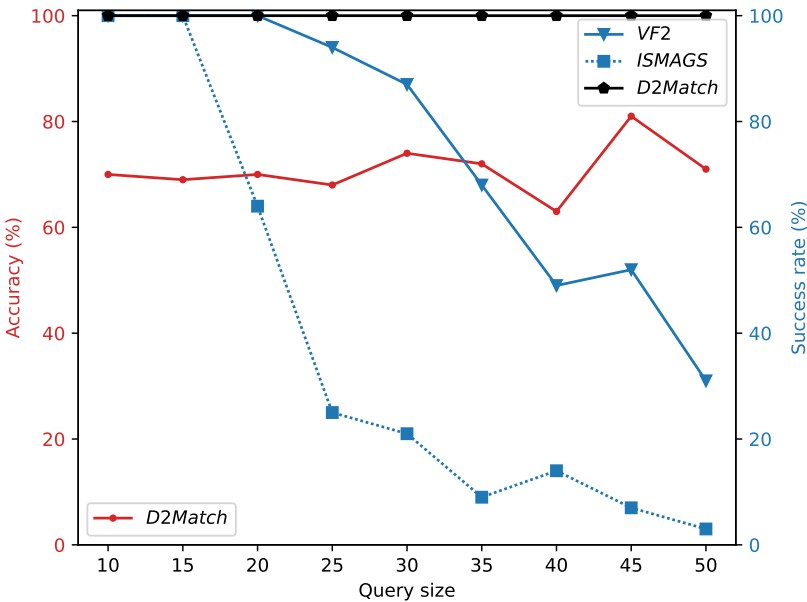

Figure 4: Comparison with exact method

