# OpenReview forum: "D2Match: Leveraging Deep Learning and Degeneracy for  Subgraph Matching"
_ICLR.cc/2023/Conference — Submitted to ICLR 2023_

### Official Review · Reviewer_763b · 2022-10-20

**Confidence:** 4
**Correctness:** 3
**Technical Novelty And Significance:** 3
**Empirical Novelty And Significance:** 3
**Recommendation:** 5

**Clarity, Quality, Novelty And Reproducibility:**

I find the proposed idea to be novel.
However, the writing of the paper should be significantly improved.
Concretely,
- The figures are not clear. For example, Figure 1 has a very brief caption, making the figure very hard to understand.
- Equation 2 is defined very vaguely. T_v^(l) is not defined, should we interpret it as a WL subtree?
- Definition of "perfect matching" is confusing, "a matching of a graph in which every node of the graph is incident to exactly one edge". Based on the definition of graph matching, we always match a query graph with a target graph. How do we match within a single graph?

**Strength And Weaknesses:**

Strength:
- The proposed idea is novel.
- The experimental results are strong.

Weakness:
- The writing of the paper should be improved.
- Although the paper provides a complexity analysis, the paper should also provide a runtime analysis to compare with baseline approaches. Given that there is an extra operation of converting a graph to a subtree, I assume the proposed method is slower compared to other deep learning based approaches. However, this is not discussed in the paper.
- Although I understand that this approach is proposed for approximate subgraph matching, I believe the paper should at least include some comparisons with exact subgraph matching methods. Please feel free to skip CO-based methods in settings where they are not applicable.
- The paper is kind of overclaiming the theoretical contribution of the paper. The paper relaxes the sufficient and necessary conditions to that of necessary only. However, I didn't find a bound or analysis regarding the error introduced in this relaxation.

**Summary Of The Paper:**

This paper identifies a degenerated case of subgraph matching, in which it falls to subtree matching. Then, the matching procedure can be implemented by the built-in tree-structured aggregation mechanism on graph neural networks. The paper proposes a new deep learning method for approximate subgraph matching.

**Summary Of The Review:**

Overall, this paper outlines a novel approach for deep learning based subgraph matching. Although I agree with the novelty of the proposed idea, I feel the paper is kind of overclaiming the theoretical contribution of the paper. The paper also needs to evaluate the proposed approach more thoroughly, by including runtime analysis and comparing against exact graph-matching baselines. Finally, the writing of the paper should be improved.
Overall, I think there is significant room for improving this paper.

---

> ### Author Response · Authors · 2022-11-14
> **Response to Reviewer 763b (1/2)**
>
> We thank Reviewer 763b for his/her insightful comments. Our responses are as follows:
>
> > Although the paper provides a complexity analysis, the paper should also provide a runtime analysis to compare with baseline approaches. Given that there is an extra operation of converting a graph to a subtree, I assume the proposed method is slower compared to other deep learning based approaches. However, this is not discussed in the paper.
> >
>
> **Answer**:
>
> - We conduct additional experiments to record the training and inference time (second) per epoch of all the comparison methods on the synthetic dataset.  Our model's scalability is acceptable because it is slower than others with a constant time. For more details, please refer to Appendix A.5.
> - Specifically, we want to highlight the following points to answer why our model is slightly slower than strong baselines.
>     1. No transformation process is needed to convert a subgraph into a subtree of the graph. We show the theoretical equivalence of using GNNs to implement subtree matching, so no explicit transformation is needed, i.e., the subtree module equals to perform GNN aggregations.
>     2. The time consumption mainly comes from the subtree module, which is linearly related to the sampling number. When we set the sampling number as 2, the running time is on par with the others. Furthermore, the running time for the GNN module is the same as for the other baselines.
>
> | Method | Training | Inference | Method | Training | Inference |
> | --- | --- | --- | --- | --- | --- |
> | SimGNN | 1.732 | 0.385 | D2Match-GNN | 2.678 | 0.495 |
> | NeuroMatch | 2.234 | 0.311 | D2Match-Subtree （S2) | 2.940 | 0.456 |
> | GraphSim | 3.223 | 0.433 | D2Match-Subtree （S3) | 3.889 | 0.581 |
> | IsoNet | 10.553 | 1.939 | D2Match-Subtree （S4) | 4.410 | 0.673 |
> | D2Match | 8.163 | 1.114 | D2Match-Subtree （S5) | 5.143 | 0.750 |
>
> > Although I understand that this approach is proposed for approximate subgraph matching, I believe the paper should at least include some comparisons with exact subgraph matching methods. Please feel free to skip CO-based methods in settings where they are not applicable.
> >
>
> **Answer**:
>
> - We agree that it is necessary to compare with the exact matching solutions.  We thus perform additional experiments showing that our model is more efficient than exact matching solutions.
> - Specifically, we use VF2[1] and ISMAGS[2] as the baselines. Since exact matching methods obtain 100% accuracy, we measure the two metrics for comparison, including the execution success rate and accuracy, where a similar comparison can be found at NeuroMatch.  By setting the execution success as the run time under the 60s, we compare the success rate of the exact methods by varying the query graph size from 10 to 50 on the synthetic data.  We list some results as follows. Please refer to Appendix A.8 for more visualized results.
>
> |  | Synthetic(20) | Synthetic(30) | Synthetic(40) | Synthetic(50) |
> | --- | --- | --- | --- | --- |
> | VF2 (success rate) | 100 | 87 | 49 | 31 |
> | ISMAGS (success rate) | 64 | 21 | 14 | 3 |
> | Ours (accuracy) | 70 | 64 | 63 | 71 |
> - We show in our experiment that the failure of exact matching methods increases significantly when the target graph has more than 30 nodes, compared to our stable performance, indicating the incompetence of these methods on large-scale datasets.
>
> > The paper is kind of overclaiming the theoretical contribution of the paper. The paper relaxes the sufficient and necessary conditions to that of necessary only. However, I didn't find a bound or analysis regarding the error introduced in this relaxation.We would like address the concern in the following.
> >
>
> **Answer**:
>
> - First, we need to emphasize that in our paper, we only claim at the bottom of the introduction, “We theoretically prove that this matching procedure can be implemented by the built-in tree-structured aggregation mechanism on GNNs and yields linear time complexity”.  Regarding this, we have proven the claim in section 4.1.
> - In terms of the bound analysis or error analysis, we appreciate the reviewer’s suggestion.  However, the bound analysis on an NP-complete problem relaxation is extremely hard [6] and is out of the scope of this paper.   We employ a necessary condition based on the degeneracy of the subgraph matching problem, a similar strategy was exploited in the literature [3][4][5].

---

> > ### Author Response · Authors · 2022-11-14
> > **Response to Reviewer 763b (2/2)**
> >
> > > The figures are not clear. For example, Figure 1 has a very brief caption, making the figure very hard to understand.
> > >
> >
> > **Answer**:
> >
> > - We added a more detailed description in Figure 1.
> >
> > > Equation 2 is defined very vaguely. T_v^(l) is not defined, should we interpret it as a WL subtree?
> > >
> >
> > **Answer**:
> >
> > - We use WL-subtree later on. In Equation 2, however, we try to make a general case to fit any function satisfying the condition. Additional details about the condition are available in Theorem 3.
> >
> > > Definition of "perfect matching" is confusing, "a matching of a graph in which every node of the graph is incident to exactly one edge". Based on the definition of graph matching, we always match a query graph with a target graph. How do we match within a single graph?
> > >
> >
> > **Answer**:
> >
> > - Note that matching has two different meanings: 1) an isomorphism between two graphs in the subgraph matching problem, and 2) a matching of two nodes happens in a single graph for the perfect matching problem. Perfect matching aims to find a set of edges such that any node exactly connects another node. More details for the definition can be found at [https://en.wikipedia.org/wiki/Perfect_matching](https://en.wikipedia.org/wiki/Perfect_matching).
> >
> > **References**
> >
> > [1] Cordella, Luigi P., et al. "A (sub) graph isomorphism algorithm for matching large graphs." *IEEE transactions on pattern analysis and machine intelligence* 26.10 (2004): 1367-1372.
> >
> > [2] Houbraken, Maarten, et al. "The Index-based Subgraph Matching Algorithm with General Symmetries (ISMAGS): exploiting symmetry for faster subgraph enumeration." *PloS one* 9.5 (2014): e97896.
> >
> > [3] Xu, Keyulu, et al. "How Powerful are Graph Neural Networks?." *International Conference on Learning Representations*. 2018.
> >
> > [4] Balcilar, Muhammet, et al. "Breaking the limits of message passing graph neural networks." *International Conference on Machine Learning*. PMLR, 2021.
> >
> > [5] Azizian, Waiss. "Expressive Power of Invariant and Equivariant Graph Neural Networks." *International Conference on Learning Representations*. 2020.
> >
> > [6] Hartmanis, Juris. "Computers and intractability: a guide to the theory of np-completeness (michael r. garey and david s. johnson)." *Siam Review* 24.1 (1982): 90

---

> ### Author Response · Authors · 2022-12-03
> **Response to Reviewer 763b**
>
> Dear Reviewer 763b,
>
> Thanks for your time and efforts in reviewing our paper. All the questions addressed in our best effort include detailed explanations, additional experiments, and writing updates. We are more than happy to discuss these questions further or any other new questions.
>
> Best Regards,
>
> Paper 545 Authors

---

> ### Author Response · Authors · 2022-12-09
> **Response to Reviewer 763b**
>
> Dear Reviewer 763b,
>
> We thank you again for your careful review.
>
> We are just sending you a friendly reminder that the discussion session is about to be over, so please let us know if you still have additional queries.
> We sincerely hope that the reviewer considers increasing the score for two reasons:
> 1) we develop a brand-new method for subgraph matching with theoretical guarantees;
> 2) we discover why previous GNN-based subgraph matching methods succeed, making them not match algorithms as claimed.
> Together, these two will deepen our understanding of the topic and lead to a better understanding of the limit of GNNs.
>
> Best,
>
> Paper 545 Authors

---

### Official Review · Reviewer_6wEv · 2022-10-22

**Confidence:** 4
**Correctness:** 3
**Technical Novelty And Significance:** 3
**Empirical Novelty And Significance:** 3
**Recommendation:** 6

**Clarity, Quality, Novelty And Reproducibility:**

The proposed idea is interesting and novel for certain types of graphs.
Writing can be improved.
1. The usage of the word “neighbor” for different scenarios at the same time made me confused. For example, in section 4: “Motivated by Hall’s marriage Theorem 3.1, we develop an efficient algorithm to address the perfect matching procedure. A straightforward solution is to randomly select a subset W from the given set of neighbors, N(q), and count whether the corresponding neighbors of W, i.e. N(W), have more elements than this subset. After repeating this process multiple times for all node pairs, we obtain a perfect matching when no instance violates the criterion.”
2. Figure 1 perfect matching table is a bit confusing.


**Strength And Weaknesses:**

Strength: Theoretical proof and support for each claims mentioned.

Weakness:
1. The datasets used in the experiments are relative small and old. It will be interesting to see the performance of the proposed method on more recent datasets, such as ogb (https://ogb.stanford.edu/)
2. Training-test split may have large impact on the performance, e.g., on Mutag and Protein. Please specify the details (e.g., random seeds, how many time the 5-fold CV was run)
3. It is unclear if the proposed method work for graphs containing edges/nodes with continuous attributes.









	 From the examples above, they have the same chordless cycle ABC and ACD


**Summary Of The Paper:**

This main contribution of this paper is that it provides an idea and corresponding theoretical proof about degenerating subgraph matching problem to subtree matching with the help of Graph Neural Network.

**Summary Of The Review:**

This study proposes an interesting method with theoretical proof. Experimental results are good. Running experiments on the suggested datasets will strengthen the paper. Writing can be improved. The paper lacks discussions about the application of the proposed method to graphs with continuous node/edge attributes.

---

> ### Author Response · Authors · 2022-11-14
> **Response to Reviewer 6wEv**
>
> We thank Reviewer 6wEv for his/her comments and the appreciation of our proposed method and corresponding proof. To the concerns expressed by Reviewer 6wEv in Weakness and Clarity, our response is as follows:
>
> > The datasets used in the experiments are relative small and old. It will be interesting to see the performance of the proposed method on more recent datasets, such as ogb ([https://ogb.stanford.edu/](https://ogb.stanford.edu/))
> >
>
> **Answer**:
>
> - We conduct additional experiments on the OGB benchmark dataset, including Ogbg-molhiv and Ogbg-molpcb, as shown in the following table. More results of these new datasets can be found in Appendix A.7.
> - Our model is slightly better than others on normal datasets while gaining a significant advantage over baselines on hard datasets. These results are consistent with our previous experiments, demonstrating that our model exploits the subgraph matching property, rather than simply modeling the divergence of the data distribution as other GNNs.
>
> | Method | Ogbg-molhiv | Ogbg-molhiv^* | Ogbg-molpcba | Ogbg-molpcba^* |
> | --- | --- | --- | --- | --- |
> | SimGNN | 99.4 | 81.6 | 99.8 | 86.2 |
> | NeuroMatch | 98.3 | 86.0 | 99.8 | 90.6 |
> | D2Match | 99.8 | 99.6 | 100.0 | 100.0 |
>
> > Training-test split may have large impact on the performance, e.g., on Mutag and Protein. Please specify the details (e.g., random seeds, how many time the 5-fold CV was run)
> >
>
> **Answer**:
>
> - We split each dataset by taking a five-fold CV similar to [1], which is based on the initial order of the raw data without any randomness.
> - We perform an additional experiment and find our model is insensitive to random seeds. We experiment on the Proteins and Mutag datasets under different random seeds and obtain nearly identical performance. Please find more details in A.4 about the effect of the randomly split.
> - As we argued in the paper (second paragraph of Section 5.3), other methods based on GNNs tend to capture the divergence of distributions in the training set, hence are easily affected by randomness. Nevertheless, by the degeneracy property, our subtree module performs the matching explicitly instead of modeling the data distribution in others, and thus ours is insensitive to data partitioning.
>
> | Seed | Proteins | Mutag |
> | --- | --- | --- |
> | 0 | 100 | 100 |
> | 1 | 100 | 100 |
> | 2 | 100 | 100 |
>
> > It is unclear if the proposed method work for graphs containing edges/nodes with continuous attributes.
> >
>
> **Answer**:
>
> - We also conduct additional experiments on continuous features from the PPI, MNIST, and CIFAR10 datasets.  Please find details of graph construction in [2] and [3].  The experimental results show that our D2Match achieves consistent performance, as shown in the following table.  Please refer to Appendix A.7 for more details.
>
> | Method | Cifar10 | MNIST | PPI |
> | --- | --- | --- | --- |
> | SimGNN | 89.0 | 98.5 | 77.0 |
> | NeuroMatch | 98.1 | 95.9 | 50.0 |
> | D2Match | 99.3 | 98.8 | 98.8 |
>
> > From the examples above, they have the same chordless cycle ABC and ACD
> >
>
> **Answer**:
>
> - We thank the reviewer for mentioning the chordless cycle because this is a critical part of our model. We are the first learning-based matching model to capture circles in a graph by using chordless cycles.  In the example, ABC (with cycle) and ACD (without cycle) differ in the cycle, but previous methods cannot capture this difference.
>
> > The usage of the word “neighbor” for different scenarios at the same time made me confused. For example, in section 4: “Motivated by Hall’s marriage Theorem 3.1, we develop an efficient algorithm to address the perfect matching procedure. A straightforward solution is to randomly select a subset W from the given set of neighbors, N(q), and count whether the corresponding neighbors of W, i.e. N(W), have more elements than this subset. After repeating this process multiple times for all node pairs, we obtain a perfect matching when no instance violates the criterion.”
> >
>
> **Answer**:
>
> - In this paper, the “neighbor” means the neighborhood of a given node or node set. In the example, N(q) is the neighbor set of node q in the original graph, and N(W) is the neighbor set of node W in the bipartite graph.  We have revised the description of this paragraph in the article.
>
> > Figure 1 perfect matching table is a bit confusing.
> >
>
> **Answer**:
>
> - The perfect matching table represents the adjacency matrix of the bipartite graph and the bolded element indicates the edges selected as a perfectly match set of edges. We added more caption for Figure 1 in the new version.
>
> **References**
>
> [1]Xu, Keyulu, et al. "How Powerful are Graph Neural Networks?." *International Conference on Learning Representations*. 2018.
>
> [2]Zitnik, Marinka, and Jure Leskovec. "Predicting multicellular function through multi-layer tissue networks." *Bioinformatics* 33.14 (2017): i190-i198.
>
> [3]Dwivedi, Vijay Prakash, et al. "Benchmarking graph neural networks." *arXiv preprint arXiv:2003.00982* (2020).

---

> ### Author Response · Authors · 2022-12-03
> **Response to Reviewer 6wEv**
>
> Dear Reviewer 6wEv,
>
> Thanks for your time and efforts in reviewing our paper.
> All the questions addressed in our best effort include detailed explanations, additional experiments, and writing updates.
> We are more than happy to discuss these questions further or any other new questions.
>
> Best Regards,
>
> Paper 545 Authors

---

### Official Review · Reviewer_9bfF · 2022-10-23

**Confidence:** 4
**Clarity, Quality, Novelty And Reproducibility:** This work may not possess good reprod…
**Correctness:** 3
**Technical Novelty And Significance:** 3
**Empirical Novelty And Significance:** 4
**Recommendation:** 6

**Strength And Weaknesses:**

**Strength:**
1. The idea of degenerating subgraph matching to subtree matching is novel. The theoretical study is interesting, regarding existing graph matching learning papers are usually empirical.
2. The proposed method improves performances on 7 experimented datasets, some by a substantial margin.
3. The empirical results show that GNNs tend to capture the data distribution divergence, and have bad performance on evenly distributed data, motivating further research.

**Weakness:**
1. The degeneracy property can be broken by simply inserting or removing an edge, so the model can be easily attacked and may not be robust to noise.
2. The ablation study of GNN shows that the improvement is mainly from the GNN module. Also, the paper does not show the subtree modeling's effect on performance improvement, such as demonstrating recall on experimented datasets.

**Summary Of The Paper:**

The authors propose a statement that subgraph matching can be degenerated to subtree matching and provide proof. Based on this degeneracy property, the authors propose a matching method that utilizes GNN to model subtrees. Also, they adopt GNN to learn node representations to boost the performance of matching.

**Summary Of The Review:**

The paper proposes a novel idea on the degeneracy property of subgraph matching, and the tree-degeneration fits the way of aggregation of GNN. The theoretical results seem novel and interesting. Empirical results show the shortcomings of GNN, which is significant for future research.

---

> ### Author Response · Authors · 2022-11-14
> **Response to Reviewer 9bfF**
>
> We thank Reviewer 9bfF for his/her comments and appreciation of our work. To the concerns expressed by Reviewer 9bfF in Weakness and Reproducibility, our response is as follows:
>
> > The degeneracy property can be broken by simply inserting or removing an edge, so the model can be easily attacked and may not be robust to noise.
> >
>
> **Answer**:
>
> - We agree that simply inserting or deleting edges can affect the subtree generation, but our model remains robust regardless of these operations.
> - Specifically, simply inserting or deleting edges are not typical noise in machine learning or deep learning, where we assume the labels remain the same when adding noise to the data. However, in our setting, even a slight modification may change the isomorphic relation, i.e., labels, contradicting the above assumption. Any good subgraph matching model will inevitably be sensitive to such noise because of the isomorphic property, including our method. As long as the isomorphism relation remains, as shown in the first paragraph of Section 4.1,”… a criterion that any isomorphic pairs can meet. ”, our D2Match is robust since the degeneracy property does not change.
>
> > The ablation study of GNN shows that the improvement is mainly from the GNN module. Also, the paper does not show the subtree modeling's effect on performance improvement, such as demonstrating recall on experimented datasets.
> >
>
> **Answer**:
>
> - We agree that the GNN module is vital, but the subtree module plays a more important role in our method.
> - To verify this, we include an ablation study and observe a clear drop in performance when removing the subtree module, as shown in the following table. Please refer to Appendix A.3 for more details.
>
> | Method | Synthetic | Synthetic+ | Proteins | Proteins^* |
> | --- | --- | --- | --- | --- |
> | D2Match (gnn only) | 61.1 | 70.2 | 95.2 | 77.2 |
> | D2Match (subtree only) | 70.0 | 74.8 | 100.0 | 82.0 |
> | D2Match | 72.7 | 86.6 | 100.0 | 83.4 |
>
> > This work may not possess good reproducibility if the code is not released.
> >
>
> **Answer**:
>
> - We release the Pytorch implementation of our model at [https://www.dropbox.com/sh/8pvj8drvj0l2zou/AAB5j7e7frVwMiun1QcCNbMFa?dl=0](https://www.dropbox.com/sh/8pvj8drvj0l2zou/AAB5j7e7frVwMiun1QcCNbMFa?dl=0).

---

> > ### Comment · Reviewer_9bfF · 2022-11-24
> > **Thanks for authors' feedback**
> >
> > I truly appreciate the authors for the feedback. This paper presents some novelty in learning subgraph matching. Yet, some issues remain not perfectly resolved: for example, the robustness w.r.t. inserting/removing an edge, which is an interesting topic if we consider the potential friendships in social networks, and it should be an advantage of ML models over traditional CO methods.
> >
> > In consideration of both the novelty and the unresolved (minor) issues, I intend to keep my original score (borderline accept).
> >
> > While I also agree with the other two reviewers that the writing of this paper needs to be further polished.

---

> > > ### Author Response · Authors · 2022-11-25
> > > **Thanks for Reviewer 9bfF's feedback**
> > >
> > > Thank you for your prompt response! We agree that the reviewer raised an interesting question and only hope to address it in future work, as this is beyond the scope of this paper.
> > > In our paper, the proposed model only considers two states: match or not, rather than similarity from a classifier ranging from 0 to 1, when performing matching. Referring to the definition of matching, inserting/removing an edge is not about robustness, as these operations change the matching condition, resulting in either matching or not matching.

---

### Decision · Program_Chairs · 2023-01-20

**Decision:**

Reject

**Justification For Why Not Higher Score:**

There is work needed on more-classical benchmarks to see how well the method does.

**Justification For Why Not Lower Score:**

N/A

**Metareview: Summary, Strengths And Weaknesses:**

This paper considers the NP-hard subgraph matching problem: given graphs G1 and G2 with G1 having at most as many vertices as G2, is there an injective map from the vertices of G1 to the vertices of G2 that exactly preserves edges? This is NP-hard since if, e.g., G1 is the complete graph on k vertices, then this subgraph matching problem is the same as the NP-hard k-clique problem on the graph G2. This paper develops some necessary conditions for this problem. It shows that subgraph matching can degenerate to subtree matching, and that the latter is equivalent to the polynomial-time-solvable perfect-matching problem on bipartite graphs. This matching procedure is further shown to be implementable by the built-in “tree-structured aggregation mechanism” of GNNs, leading to a linear time complexity in some cases. It is good to see such ML-based approaches to hard CO problems.

Some of the concerns raised include robustness with respect to the addition or deletion of graph edges, which needs further study. Here is an important classical benchmark problem due to Richard Karp, for the authors to try their algorithms on (and to possibly improve their algorithms). Generate the Erdos-Renyi random graph G = G(n, ½)---i.e., G has n vertices and an edge is added between each pair of edges independently with probability ½. With high probability, the maximum clique size in this graph is around 2 * log_2 n. It is a very interesting problem to construct even a clique of size, say, 1.5 * log_2 n in such a graph. That is, the authors are encouraged to try their method when G1 is the complete graph on k vertices where k = floor(1.5 * log_2 n), and G2 is the random graph G(n, ½) where n is, say, on the order of several thousands.

The authors are also asked to improve their writing as suggested by the referees.


**Summary Of Ac-Reviewer Meeting:**

N/A